# Jacta: A Versatile Planner for Learning Dexterous and Whole-body Manipulation

**Jan Brüdigam**[1], **Ali-Adeeb Abbas**[2], **Maks Sorokin**[2], **Kuan Fang**[2], **Brandon Hung**[2],
**Maya Guru**[2], **Stefan Sosnowski**[1], **Jiuguang Wang**[2], **Sandra Hirche**[1], **Simon Le Cleac'h**[2]

TU Munich[1], Boston Dynamics AI Institute[2]

**Abstract:** Robotic manipulation is challenging due to discontinuous dynamics, as well as high-dimensional state and action spaces. Data-driven approaches that succeed in manipulation tasks require large amounts of data and expert demonstrations, typically from humans. Existing planners are restricted to specific systems and often depend on specialized algorithms for using demonstrations. Therefore, we introduce a flexible motion planner tailored to dexterous and whole-body manipulation tasks. Our planner creates readily usable demonstrations for reinforcement learning algorithms, eliminating the need for additional training pipeline complexities. With this approach, we can efficiently learn policies for complex manipulation tasks, where traditional reinforcement learning alone only makes little progress. Furthermore, we demonstrate that learned policies are transferable to real robotic systems for solving complex dexterous manipulation tasks.

Project website: https://jacta-manipulation.github.io/

**Keywords:** Dexterous Manipulation Planning, Learning with Demonstrations

## 1 Introduction

Dexterous and whole-body robotic manipulation in high-dimensional state-action space with non-smooth dynamics is challenging, but learning-based approaches have shown encouraging results for such tasks. However, they require loads of high-quality data [1, 2] or extensive exploration to find solutions. Yet, robot data is scarce and expensive; human demonstrations are costly and confined to human capabilities. Moreover, using dense rewards or curriculum learning to guide the policy search potentially imposes a sub-optimal solution structure and necessitates lengthy reward shaping.

A promising solution to these issues is to leverage motion planners to automatically generate demonstrations for learning instead of relying on humans [3, 4, 5]. This approach addresses the challenge of scaling data collection. However, existing manipulation planners are often designed for specific end effectors and single tasks such as pushing [6, 7], pick-and-place, [8, 9, 10] or in-hand manipulation [11, 12]. Further, using the demonstrations generated by these motion planners is non-trivial because they assume quasi-static dynamics [13, 14, 15] or only explicitly provide state trajectories [12]. This paper shows that robotic manipulation must not be limited in this manner. A planner can combine highly dexterous capabilities to precisely manipulate small items and whole-body interactions to maneuver large objects, with contact occurring not just at the end effector.

To address these issues, we introduce a novel planner and learning pipeline for dexterous and whole-body robotic manipulation, shown in Fig. 1. The planner finds solutions to manipulation tasks through sampling-based and gradient-based actions for global and goal-directed exploration. It is not limited to quasi-static systems, nor does it need to explicitly reason about the contact modes. The aim is not to use the planner directly in a receding horizon fashion akin to model predictive control but to leverage planner-generated demonstrations to bootstrap policy learning. Thus, when using the planner, one is not bound to human demonstrations and human-like robotic systems. Instead, one

8th Conference on Robot Learning (CoRL 2024), Munich, Germany.

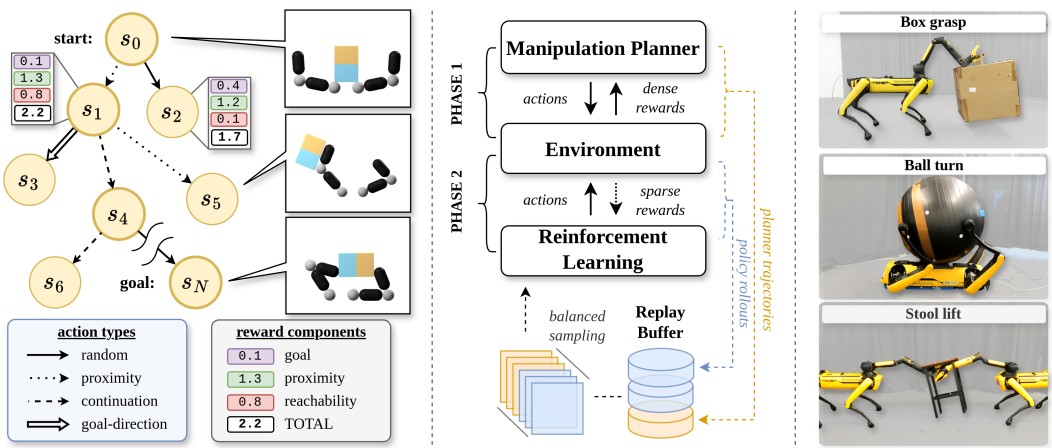

Figure 1: Our manipulation planner generates a search tree of dynamically feasible state-action trajectories. During simulation training, planner trajectories are fed to the replay buffer of a reinforcement learning algorithm. The resulting simulation-trained policies are deployed on real robotic systems for different manipulation tasks.

can build on learning methods that are bootstrapped with the planner's solutions. Surprisingly, we find that simply adding a small number of planner demonstrations to the replay buffer is enough to drastically improve learning performance and generate successful policies for complex tasks.

In summary, our contributions are the following:

- Design of a novel motion planner generating dynamically feasible state-action trajectories for dexterous and whole-body manipulation tasks.
- Identification of a simple yet efficient learning pipeline for using the planner's demonstrations in reinforcement learning.
- Transfer of trained policies to real systems to demonstrate the feasibility of complex tasks covering both dexterous and whole-body manipulation.

## 2 Related work

**Guiding Manipulation Learning with Demonstrations**   Data-driven approaches are state-of-the-art for solving complex manipulation tasks. Imitation learning (IL) leverages expert demonstrations of such tasks to train policies for robots. Recent results show the great potential of IL in advanced tasks, such as unseen object manipulation [16, 17], door opening [18], or manipulation of liquids and fabrics [19]. Yet, IL requires large amounts of expert demonstrations to deliver working policies, which is a bottleneck for quickly learning new skills, especially since human demonstrations are inherently limited to human capabilities and costly to collect. Reinforcement learning (RL) does not necessarily require demonstrations due to its noisy exploration. However, the initially random exploration in large state-action spaces has a low chance of success. Accordingly, demonstrations have also been used to guide RL with additional offline data. Two different ways of incorporating demonstrations into RL have found popularity: directly adding the demonstrations to the replay buffer with minor modifications to the underlying learning algorithm [20, 21, 22] or pre-training an imitation policy from the demonstrations [23, 24, 25]. Still, the advanced manipulation tasks investigated in [23, 24] only cover anthropomorphic systems and require human demonstrations. In contrast, we propose a pipeline with a planner that is not limited to anthropomorphic systems and eliminates the need for costly human data collection, thus offering greater scalability.

**Planning for Robotic Manipulation Learning**   While robotic motion planning has seen great success, especially with sampling-based methods [26, 27], manipulation planning remains particularly difficult due to the multitude of contacts in general settings. Thus, most methods focus on motion planning for low-dexterity grippers and specialized tasks such as pushing or fixed grasping

[6, 7, 8, 28, 29]. More advanced techniques plan dexterous manipulation assuming quasi-static contact dynamics [15, 14] and myopic sampling-based planning offers compelling results but only on short-horizon tasks [30]. Manipulation planning based on optimal control often requires smoothed contact dynamics [31] or scheduled contact sequences [32, 33] since contact-implicit schemes are limited to simple tasks in practice [34, 35, 36]. In contrast, our proposed planner is designed to solve dexterous and whole-body robotic manipulation tasks without restrictions to specific end effectors or contact locations. We allow contact on all parts of a robot and plan with full contact dynamics to generate readily usable demonstrations for manipulation learning.

Since manipulation planners are not fast enough to be directly deployed on the robot systems, they have been combined with learning methods. Collision-free motion planners have been used as an alternative to RL's random exploration, but the applications have been limited to simple manipulation tasks [4, 3]. Moreover, motion-planner actions do not overcome the lack of reward signals in sparse reward settings [37, 38]. Demonstrations from motion planners have served as expert data for IL with pick-and-place tasks [5] and for RL with basic reaching tasks rather than full-scale manipulation [39, 40]. Planner-based demonstrations for manipulation have been used to find an informed start-state distribution for the RL exploration for simple object rearrangement tasks [13], and a similar approach has been developed for dexterous in-hand manipulation [12]. However, these methods implement task-specialized planners and are not directly portable to generic manipulation settings. Additionally, they only provide state trajectories without associated actions, making it difficult to use the demonstrations directly for RL. Both of these issues are addressed with our planner, which produces dynamically feasible state-action demonstrations for general robotic manipulation tasks.

## 3 Background

This section describes the robotic manipulation scenario solved by our motion planner and the reinforcement learning algorithm used to optimize policies for the manipulation tasks.

**Robotic manipulation scenario** We consider one or multiple robots with joint configuration $q_{\mathrm{r}} \in \mathbb{R}^{n_{\mathrm{r}}}$ and joint velocity[1] $\dot{q}_{\mathrm{r}} \in \mathbb{R}^{n_{\mathrm{r}}}$ that interact with one or multiple objects with configuration $q_{\mathrm{o}} \in \mathbb{R}^{n_{\mathrm{o}}}$ and velocity $\dot{q}_{\mathrm{o}} \in \mathbb{R}^{n_{\mathrm{o}}}$. These quantities form the state $s = [q_{\mathrm{r}}^{\mathsf{T}} \ \dot{q}_{\mathrm{r}}^{\mathsf{T}} \ q_{\mathrm{o}}^{\mathsf{T}} \ \dot{q}_{\mathrm{o}}^{\mathsf{T}}]^{\mathsf{T}} \in \mathbb{R}^{n_{\mathrm{s}}}$. The actions $a \in \mathbb{R}^{n_{\mathrm{r}}}$ are position commands for the robots' actuated joints. The position commands $a$ are tracked by a low-level controller (cf. Appendix A). The potentially non-smooth dynamics for this system are $s_{t+1} = f(s_t, a_t)$ at time $t$ and contain bounds on the robots' and objects' states. The planner's task is to find a state-action trajectory that minimizes the weighted distance $d(s, s_{\mathrm{g}}) = \|s - s_{\mathrm{g}}\|_{Q_{\mathrm{d}}}$ to a desired goal state $s_{\mathrm{g}}$. The weight matrix $Q_{\mathrm{d}} \in \mathbb{R}^{n_{\mathrm{s}} \times n_{\mathrm{s}}}$ is used to prioritize certain dimensions of the state, e.g., of the objects, and ignore others, e.g., of the robots.

**Reinforcement learning algorithm** Manipulation policies are found with value-based reinforcement learning [41, 42]. A goal-oriented value function $Q(s, a, s_{\mathrm{g}}) \in \mathbb{R}$ for the system's state-action-goal space is learned from rewards $r$. The sparse reward function $r(\cdot) \in \{-1, 0\}$ is 0 if the relative distance $d$ of a state $s_t$ to the start state $s_0$ is below a threshold $\epsilon$, and $-1$ otherwise:

$$r(s_t, s_{\mathrm{g}}) = \begin{cases} 0 & \text{if } \frac{d(s_t, s_{\mathrm{g}})}{d(s_0, s_{\mathrm{g}})} \leq \epsilon \\ -1 & \text{otherwise} \end{cases}. \tag{1}$$

The optimal value function given this reward function is

$$Q^{\star}(s_t, a_t, s_{\mathrm{g}}) = \mathbb{E}_{s_{t+1}} \left[ r(s_t, s_{\mathrm{g}}) + \gamma \max_{a_{t+1}} Q^{\star}(s_{t+1}, a_{t+1}, s_{\mathrm{g}}) \right], \tag{2}$$

where $\gamma \in [0, 1)$ is a discount factor. Approximating the value function with a network $Q_{\phi}$ enables learning the network's parameters $\phi$ from tuples $(s_t, a_t, s_{\mathrm{g}}, r_t, s_{t+1})$ in continuous spaces. The optimal policy $a = \pi^{\star}(s, s_{\mathrm{g}})$ maximizes the value function:

$$\pi^{\star}(s, s_{\mathrm{g}}) = \arg\max_{a} Q^{\star}(s, a, s_{\mathrm{g}}), \tag{3}$$

---

[1]If the robot state contains a quaternion, the joint velocity is lower dimensional than the joint configuration.

and is also approximated by a network $\pi_{\boldsymbol{\theta}}$ with parameters $\boldsymbol{\theta}$, which provides a continuous policy representation. Since we want to use demonstrations from our motion planner for learning, we use the off-policy learning algorithm deep deterministic policy gradients (DDPG) [41].

## 4 Dexterous and whole-body manipulation planner

We design a motion planner to generate dynamically feasible state-action trajectories for dexterous and whole-body manipulation. These solutions serve as demonstrations to bootstrap policy learning. The planner builds a tree starting from a single start state by taking actions towards the goal. For each action taken, a new node is added to the tree, as shown in Fig. 1. In this way, the search space is explored until a viable trajectory to the goal is found.

---
**Algorithm 1** Manipulation Planner
---
1: Choose $n_{\mathrm{g}}, n_{\mathrm{i}}, \beta, n_{\mathrm{e}}, p_{\mathrm{a}}, \Delta t, \boldsymbol{s}_1$; Set $n_{\mathrm{n}} = 1$
2: **for** $i_{\mathrm{g}}$ in $1 : n_{\mathrm{g}}$ **do**
3:    $\boldsymbol{s}_{\mathrm{g}} = $ goal_sampling$(b_{\mathrm{g}}, \boldsymbol{s}_{\mathrm{min}}, \boldsymbol{s}_{\mathrm{max}})$ ▷ Sec. 4.1
4:    reward_update$(\boldsymbol{s}_{\mathrm{g}})$          ▷ Sec. 4.2
5:    **for** $i_{\mathrm{i}}$ in $1 : n_{\mathrm{i}}$ **do**
6:      $i_{\mathrm{n}}, n_{\mathrm{e}} = $ node_selection$(\beta, n_{\mathrm{e}})$    ▷ Sec. 4.3
7:      **for** $i_{\mathrm{e}}$ in $1 : n_{\mathrm{e}}$ **do**
8:        $\boldsymbol{a}_{n_{\mathrm{n}}+1} = $ action_sampling$(p_{\mathrm{a}})$    ▷ Sec. 4.4
9:        $\boldsymbol{s}_{n_{\mathrm{n}}+1} = $ extension$(\boldsymbol{s}_{n_{\mathrm{n}}}, \boldsymbol{a}_{n_{\mathrm{n}}+1})$ ▷ Sec. 4.5
10:        $n_{\mathrm{n}} = n_{\mathrm{n}} + 1$
11:      $i_{\mathrm{n}} = n_{\mathrm{n}}$
---

Robotic manipulation has discontinuous contact dynamics and rugged optimization landscapes [43]. Thus, the planner involves several components. Gradient-based actions for locally optimal search are combined with random actions for global exploration and avoiding local minima. Reward heuristics are used to make progress in high-dimensional spaces. The resulting planner is applicable to general robots and manipulation tasks. We only assume the availability of a simulator for dynamics and gradients, e.g., Mujoco [44]. The search algorithm is stated in Alg. 1, and the components of the planner are introduced in the subsections below. Ablations for the planner's parameters are given in Appendix C.

### 4.1 Goal sampling

Manipulation settings have highly constrained non-convex search spaces with many local minima. Inspired by rapidly exploring random trees [26], $n_{\mathrm{g}}$ different goal states are sampled during the search to escape local minima and enable global exploration. A bias $b_{\mathrm{g}} \in [0, 1]$ is set to sample with a certain probability the actual goal for a given task $\boldsymbol{s}_{\mathrm{task}}$. Otherwise, a random state is sampled uniformly in a box $\boldsymbol{s}_{\mathrm{random}} \in [\boldsymbol{s}_{\mathrm{min}}, \boldsymbol{s}_{\mathrm{max}}]$ with upper and lower bounds $\boldsymbol{s}_{\mathrm{min}}$ and $\boldsymbol{s}_{\mathrm{max}}$:

$$\boldsymbol{s}_{\mathrm{g}} = \begin{cases} \boldsymbol{s}_{\mathrm{task}} & \text{if } p_{\mathrm{g}} \leq b_{\mathrm{g}} \\ \boldsymbol{s}_{\mathrm{random}} & \text{otherwise} \end{cases}, \quad \text{with } p_{\mathrm{g}} \sim \mathcal{U}(0, 1), \ \boldsymbol{s}_{\mathrm{random}} \sim \mathcal{U}(\boldsymbol{s}_{\mathrm{min}}, \boldsymbol{s}_{\mathrm{max}}). \quad (4)$$

Sampled non-unit-length orientation representations, i.e., quaternions, are normalized. Infeasible states, i.e., states with contact penetration, are discarded and resampled.

### 4.2 Reward metric

Exploration in high-dimensional spaces is challenging due to the combinatorial explosion. Thus, each node in the tree is assigned a value to permit a heuristic selection of promising nodes during the search, similar to Monte Carlo tree search [45]. Instead of a discounted value, directly using a node's reward yields satisfactory results for us and removes the need for value back-propagation up the tree. When the goal state $\boldsymbol{s}_{\mathrm{g}}$ is changed, the total reward (value) of a node with state $\boldsymbol{s}$,

$$r(\boldsymbol{s}, \boldsymbol{s}_{\mathrm{g}}) = r_{\mathrm{d}}(\boldsymbol{s}, \boldsymbol{s}_{\mathrm{g}}) + r_{\mathrm{p}}(\boldsymbol{s}) + r_{\mathrm{m}}(\boldsymbol{s}, \boldsymbol{s}_{\mathrm{g}}), \quad (5)$$

is computed from the three following reward components.

**Distance** The actual closeness to the desired manipulation goal is formulated as a distance reward,

$$r_{\mathrm{d}}(\boldsymbol{s}, \boldsymbol{s}_{\mathrm{g}}) = - \left\| \boldsymbol{s} - \boldsymbol{s}_{\mathrm{g}} \right\|_{\boldsymbol{Q}_{\mathrm{d}}}, \quad (6)$$

with weight matrix $\boldsymbol{Q}_{\mathrm{d}} \in \mathbb{R}^{n_{\mathrm{s}} \times n_{\mathrm{s}}}$. It encourages a small-scaled distance between a node's state and the goal. Note that for quaternions, the three-dimensional orientation difference is used [46].

**Proximity**  For manipulation, robot-object interactions must occur. Such interactions are more likely if the robot is close to the object and a reward is given for such states. The distance between robots and objects is measured by $n_\mathrm{p}$ virtual proximity sensors. Sensor $i \in \{0, \ldots, n_\mathrm{p}\}$ consists of a pair of points with $\boldsymbol{p}_{\mathrm{r},i}$ attached to the robot hardware and $\boldsymbol{p}_{\mathrm{o},i}$ attached to the object. It returns a distance $d_i = \|\boldsymbol{p}_{\mathrm{r},i} - \boldsymbol{p}_{\mathrm{o},i}\|$. The proximity reward for a node is

$$r_\mathrm{p}(\boldsymbol{d}) = - \|\boldsymbol{d}\|_{\boldsymbol{Q}_\mathrm{p}}, \tag{7}$$

where $\boldsymbol{Q}_\mathrm{p} \in \mathbb{R}^{n_\mathrm{p} \times n_\mathrm{p}}$ is a weight matrix and $\boldsymbol{d} = [d_1, \ldots, d_{n_\mathrm{p}}]^\mathsf{T} \in \mathbb{R}^{n_\mathrm{p}}$ consists of all stacked sensor values. Close proximity between the robot and the object ensures frequent whole-body interactions but may prevent exploration because moving away from the object to change the robot's configuration is penalized. The weight matrix $\boldsymbol{Q}_\mathrm{p}$ is responsible for achieving a good trade-off.

**Reachability**  When being close to an object, a robot needs to be in a configuration where moving the object towards the goal is easily possible. Given an object's state difference from the goal $\Delta \boldsymbol{s}_\mathrm{o} = \boldsymbol{s}_\mathrm{o} - \boldsymbol{s}_{\mathrm{g},\mathrm{o}}$, a measure $m$ for the difficulty of reaching the goal can be calculated with the local Mahalanobis metric defined in [14] as

$$m = \Delta \boldsymbol{s}_\mathrm{o}^\mathsf{T} \boldsymbol{M} \Delta \boldsymbol{s}_\mathrm{o}, \quad \text{with } \boldsymbol{M} = \left(\boldsymbol{B}_\mathrm{o} \boldsymbol{B}_\mathrm{o}^\mathsf{T} + \mu \boldsymbol{I}\right)^{-1} \in \mathbb{R}^{n_\mathrm{o} \times n_\mathrm{o}}, \ \boldsymbol{B}_\mathrm{o} = \frac{\partial \boldsymbol{f}_\mathrm{o}}{\partial \boldsymbol{a}} \in \mathbb{R}^{n_\mathrm{o} \times n_\mathrm{a}}. \tag{8}$$

Here, $\boldsymbol{B}_\mathrm{o}$ is the control Jacobian of the object dynamics $\boldsymbol{f}_\mathrm{o}$ with respect to the action $\boldsymbol{a}$, $\mu \in \mathbb{R}$ is a regularization parameter to ensure the invertibility of $\boldsymbol{M}$, and $\boldsymbol{I} \in \mathbb{R}^{n_\mathrm{o} \times n_\mathrm{o}}$ is the identity matrix. A small $m$ indicates that moving the object towards the goal is easy in the current robot configuration. The reachability reward based on $m$ is defined as

$$r_\mathrm{m}(m) = -q_\mathrm{m} \log(m), \tag{9}$$

with weight $q_\mathrm{m}$. In practice, $m$ can be clipped to avoid overly large rewards (cf. Appendix A).

### 4.3   Node selection and extension horizon

At the beginning of each of the $n_\mathrm{i}$ iterations, a node with index $i_\mathrm{n} \in \mathbb{N}_+$ must be selected. A trade-off is required between selecting the currently best node for greedy exploitation and selecting worse nodes for exploration. To this end, the tree's $n_\mathrm{n}$ nodes are ranked from highest to lowest reward, and the index $i_\mathrm{n} = [x \sim \mathcal{P}(n_\mathrm{n}, \beta)]$ is the rounded sample from a truncated Pareto distribution [47] with bounds $[1, n_\mathrm{n}]$ and exponent $\beta \in \mathbb{R}_+$. Larger values for the parameter $\beta$ lead to a more greedy node selection. After the first node is selected, given the extension horizon $n_\mathrm{e}$, the remaining $n_\mathrm{e} - 1$ node indices for the current iteration are the last extended nodes' indices. In practice, $\beta$ and $n_\mathrm{e}$ can be varied when the search is progressing well or poorly (cf. Appendix A).

### 4.4   Actions

Exploration is based on a mix of globally exploring random actions to avoid local minima and locally optimal actions for precise fine-grained manipulation. Once a node is selected, a step size $\Delta t_\mathrm{a}$ and an action $\boldsymbol{a} = \hat{\boldsymbol{a}} \circ \boldsymbol{\alpha}$ is generated with element-wise multiplication of the direction $\hat{\boldsymbol{a}} \in \{\mathbb{R}^{n_\mathrm{a}} \mid \|\hat{\boldsymbol{a}}\| = 1\}$ and magnitude $\boldsymbol{\alpha} \in \{\mathbb{R}^{n_\mathrm{a}} \mid \boldsymbol{0} \leq \boldsymbol{\alpha} \leq \boldsymbol{\alpha}_\mathrm{max}\}$, where $\boldsymbol{\alpha}_\mathrm{max}$ is the maximum magnitude. The action corresponds to a relative position command and is calculated using one of the methods listed below. The method used is sampled according to a user-specified discrete probability distribution $p_\mathrm{a}$. In practice, the step size $\Delta t_\mathrm{a}$ for each action can vary (cf. Appendix A).

**Random**  To ensure global exploration, a random action can be taken. This action samples a uniformly distributed direction $\hat{\boldsymbol{a}}$, and the magnitude $\boldsymbol{\alpha}$ is sampled uniformly within the magnitude bounds.

**Continuation**  Obtaining a promising search direction is challenging, so instead of discarding a successful action after one step, it can prove beneficial to keep going in this direction. Accordingly, the continuation action takes the same action direction that leads to the currently selected node, i.e., $\hat{\boldsymbol{a}}_{i_\mathrm{n}} = \hat{\boldsymbol{a}}_{j_\mathrm{n}}$, where $i_\mathrm{n}$ and $j_\mathrm{n}$ are the current and parent node indices, respectively. The magnitude $\boldsymbol{\alpha}$ is sampled uniformly within the magnitude bounds.

**Proximity**   As explained for the proximity reward, closeness of robot and object is necessary for manipulation. Therefore, the values $\boldsymbol{d}$ of the virtual proximity sensors are also used to compute the proximity action. The action direction aims to minimize $\frac{1}{2}\boldsymbol{d}^{\mathsf{T}}\boldsymbol{d}$ and points in the opposite direction of the gradient $\boldsymbol{g} = (\partial\boldsymbol{d}/\partial\boldsymbol{a})^{\mathsf{T}}\boldsymbol{d}$, i.e., $\hat{\boldsymbol{a}} = -\boldsymbol{g}/\|\boldsymbol{g}\|$. The magnitude $\boldsymbol{\alpha}$ is sampled uniformly within the magnitude bounds.

**Goal-direction**   Besides random actions for global exploration, goal-directed actions are required to make meaningful progress. Locally-optimized actions are computed by linearizing the dynamics around the reference values $\boldsymbol{s}_0^* = \boldsymbol{s}_0$ and $\boldsymbol{a}_0^* = \boldsymbol{a}_0 - \Delta\boldsymbol{a}_0$, with the fixed reference state $\boldsymbol{s}_0$:

$$\boldsymbol{f}(\boldsymbol{s}_0, \boldsymbol{a}_0) \approx \boldsymbol{f}_0^* + \boldsymbol{B}\Delta\boldsymbol{a}_0, \quad \text{with } \boldsymbol{f}_0^* = \boldsymbol{f}(\boldsymbol{s}_0^*, \boldsymbol{a}_0^*), \ \boldsymbol{B} = \left.\frac{\partial\boldsymbol{f}}{\partial\boldsymbol{a}}\right|_{\boldsymbol{s}_0^*, \boldsymbol{a}_0^*} \in \mathbb{R}^{n\times m}. \tag{10}$$

The linearized dynamics are used to solve the one-step optimal control problem

$$\min_{\Delta\boldsymbol{a}_0} \frac{1}{2}\left(\boldsymbol{s}_1 - \boldsymbol{s}_\mathrm{g}\right)^{\mathsf{T}}\boldsymbol{Q}\left(\boldsymbol{s}_1 - \boldsymbol{s}_\mathrm{g}\right) + \frac{1}{2}\left(\boldsymbol{a}_0^* + \Delta\boldsymbol{a}_0\right)^{\mathsf{T}}\boldsymbol{R}\left(\boldsymbol{a}_0^* + \Delta\boldsymbol{a}_0\right) \tag{11a}$$

$$\text{s.t.} \quad \boldsymbol{s}_1 = \boldsymbol{f}_0^* + \boldsymbol{B}\Delta\boldsymbol{a}_0, \tag{11b}$$

with a closed-form solution for $\Delta\boldsymbol{a}_0$ (cf. Appendix A). The resulting action direction is $\hat{\boldsymbol{a}} = (\boldsymbol{a}_0^* + \Delta\boldsymbol{a}_0)/\|\boldsymbol{a}_0^* + \Delta\boldsymbol{a}_0\|$, where $\boldsymbol{a}_0^*$ can be chosen as, for example, the action used to reach the current node. The action magnitude is set to the maximum value, i.e., $\boldsymbol{\alpha} = \boldsymbol{\alpha}_{\max}$.

## 4.5   Tree extension

Once the action is computed for a selected node, a dynamics rollout is performed. The reached state is appended to the search tree as a new node. Details on the rollout are given in Appendix A.

## 5   Bootstrapping reinforcement learning with demonstrations

Our planner can find solutions for challenging manipulation tasks but provides demonstrations only for a single start state. Accordingly, we do not deploy the planner directly as an online policy but instead use the demonstrations to bootstrap reinforcement Learning (RL). This approach allows us to generalize beyond the planner's demonstrations and perform additional online exploration to exceed the planner's suboptimal performance stemming from partially random actions. By distilling a policy, we also no longer require the planner during policy training or execution. For the best inclusion of demonstration, we investigate several different methods, which can be grouped into the two main strategies outlined below. Further details are provided in Appendix B. Foreshadowing the results, we find that using a fixed ratio of demonstrations in the replay buffer provides the best performance of all investigated methods. This result aligns with the core idea of off-policy RL [48] and findings in [22]. It is a welcoming result due to the simplicity of implementing this strategy.

## 5.1   Filling the replay buffer with demonstrations

During each training epoch, $n_\mathrm{r} = n_\mathrm{p} + n_\pi$ rollouts, consisting of $n_\mathrm{p}$ offline planner demonstrations and $n_\pi$ online policy rollouts, are appended to the replay buffer. Afterward, training data is sampled randomly from the entire replay buffer. Similar to [22], we investigate three different approaches for filling the replay buffer: (1) using a fixed ratio $n_\mathrm{p}/n_\pi$ in each epoch, (2) using a decreasing ratio as the policy's success rate increases, and (3) adding only planner demonstrations in the first epoch and only online policy rollouts afterward.

## 5.2   Pre-training policies and value functions

Following the concepts outlined in [25], we investigate three different approaches for using demonstrations for pre-training: (1) pre-training an imitation policy $\pi_\mathrm{IL}$ from demonstrations and using it as a fixed policy alongside the RL-trained policy $\pi_\mathrm{RL}$, (2) pre-training $\pi_\mathrm{IL}$ and initializing the value function $Q$ from demonstrations, and (3) pre-training $\pi_\mathrm{IL}$ as well as initializing $Q$ and additionally adding a fixed ratio of demonstrations to the replay buffer as described above.

# 6 Experiments

Given our motion planning and learning setup, we investigate three main questions:

A) Can the planner generate solutions for dexterous and whole-body manipulation tasks?

B) Can the solution demonstrations bootstrap manipulation learning?

C) Can the learned policies be transferred to real systems?

These questions are evaluated for the six tasks displayed in Fig. 1 and 2. Note that additional tasks, baseline comparisons, and ablations, as well as details on systems, parameters, and experiment setups, are given in Appendix C. For each evaluation run, five random seeds are used.

**A) Manipulation planning** The planner's ability to solve the manipulation tasks is measured as the search progress $= 1 - r_d(s_t, s_g)/r_d(s_0, s_g)$, i.e., the scaled relative closeness to the goal (cf. (6)). This progress is plotted for the six tasks in Fig. 2. Despite the varying tasks and complexity of the settings, the planner consistently finds solutions for all scenarios, demonstrating that the rewards and actions are general enough to treat high-dimensional spaces and multi-contact dynamics.

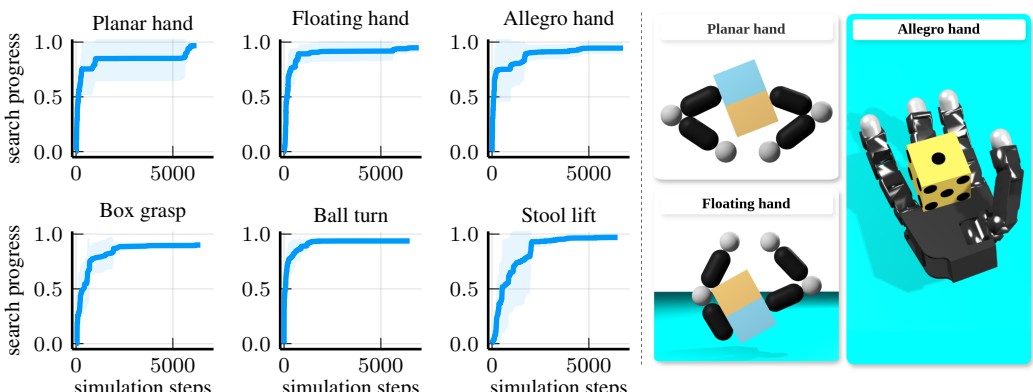

Figure 2: **Left**: the performance of the Manipulation Planner (Alg. 1) on six tasks. Relative closeness to the goal for the best node in the tree so far vs. the total number of nodes (simulation steps). **Right**: the three additional tasks used for evaluation besides the tasks shown in Fig. 1.

**B) Learning with Demonstrations** We compare the following approaches for demonstration inclusion in Fig. 3: pure DDPG as a baseline, using a fixed ratio of $n_p/n_r = 25\%$ demonstrations for the replay buffer (cf. Sec. 5.1), and pre-training an imitation policy as a secondary policy (cf. Sec. 5.2). For comparability with the planner, the learning progress is plotted over the total actions taken. Using demonstrations always leads to an improvement over pure reinforcement learning, sometimes drastically, while pre-training an imitation policy performs inconsistently. Even though the demonstrations are limited to a single start state, they provide enough reward signals to learn successful policies faster and more robustly for all runs. At the same time, the demonstrations are not good enough to directly train and use an imitation policy, as adding this policy does not yield satisfactory results as it is slower and does not make any learning progress for some of the runs.

**C) Transfer to real systems** We transfer policies from training in simulation directly to the hardware for the three tasks shown in Fig. 1. For each setting, we pick a policy with good performance in simulation but perform no additional training in the real world. Table 1 shows the evaluation success rates on an initial-state grid, and Fig. 4 shows the progression of the stool-lift task as an example.

|           | Trials | Success |
|-----------|--------|---------|
| Box grasp | 27     | 93%     |
| Ball turn | 20     | 50%     |
| Stool lift | 25    | 76%     |

Table 1: Success rate on real systems.

The trained policies for the box-grasp and stool-lift tasks work well, given the direct policy deployment. It is worth noting that in the stool-lift task, contact occurs on the robot arms, not just the

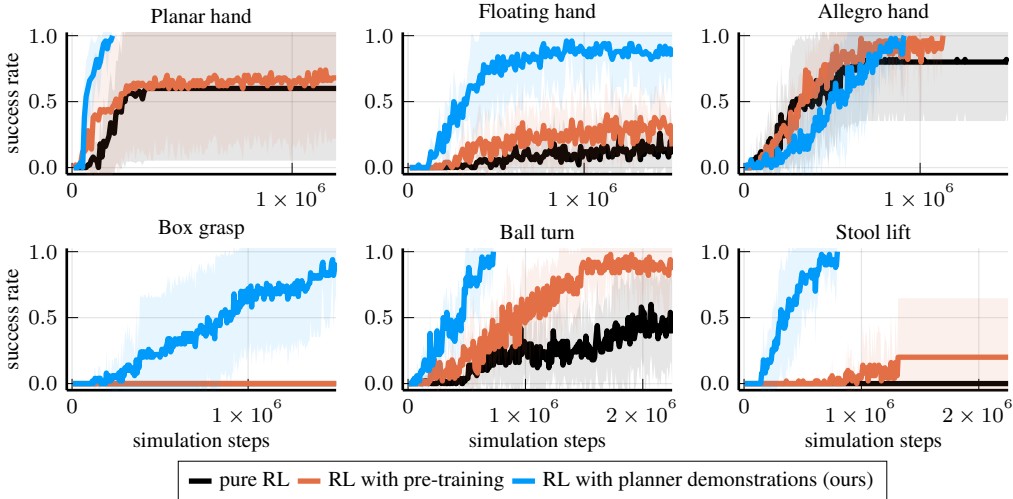

Figure 3: The performance of reinforcement learning for manipulation on six tasks. Comparison of our approach of adding demonstrations to the replay buffer (blue), RL with an additional imitation policy pre-training (red), and pure RL (black). Further baseline comparisons are in Appendix C.

end effectors. Transferring the ball-turn task to the real world is more challenging due to modeling approximations, as the yoga ball is simulated as a perfectly rigid object.

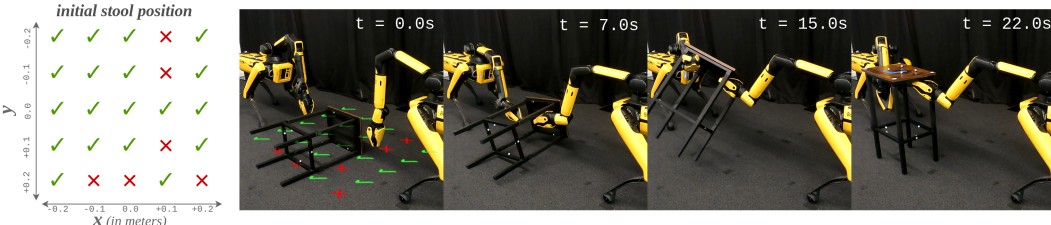

Figure 4: **Left**: the successful and failed initial positions for the stool-lift tasks. **Right**: deployment of the trained policy for two Spots lifting and placing a stool in the upright position.

## 7 Conclusions

We present a versatile planner for dexterous and whole-body manipulation tasks. The planner generates demonstrations that are used to bootstrap reinforcement learning, leading to significant learning improvements. It is robust against increasing task complexity, making it a valuable tool for general and complex manipulation scenarios. Similarly, aiding reinforcement learning with synthetic demonstrations from the planner is simple to implement yet gives strong results, providing a promising architecture choice for general manipulation learning, also in combination with other methods.

**Limitations** Our method has limitations. (1) Like any automated search, the planner requires hyperparameter tuning. However, the planner can provide results within seconds compared to learning algorithms that often require minutes or hours before success can be assessed. This fast process drastically reduces the iteration loop and allows manual tuning at interactive rates and quick parameter sweeps. (2) Both the planner and policies rely on state information, which requires a motion capture system in the real world to acquire such state information. There are several ways to extend this work to synthesize vision-based policies. One could transfer planner data to convert state-action trajectories to image-action trajectories. A second option would be to apply a teacher-student framework [49, 50, 51, 52, 38] by training a student policy that only has access to vision sensors to mimic our expert state-based policy. (3) Finally, we use a relatively simple pre-training setup, and the reported performance of imitation pre-training could change with more sophisticated IL methods.

**Acknowledgments**

We would like to thank Emmanuel Panov, Ciarán T. O'Neill, and Stephen Proulx for their help with the hardware experiments. We would also like to thank Dmitry Yershov, Gustavo Nunes Goretkin, Farzad Niroui, Martin Schuck, and Petar Bevanda for technical discussions and implementation help. Finally, we would like to thank Tao Pang and Tong Zhao for their help and guidance in running baseline benchmark comparisons.

This work was supported by the European Union's Horizon Europe innovation action programme under grant agreement No. 101093822, "SeaClear2.0".

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

# A   Manipulation Planner implementation details

Implementation details and parameters for the manipulation planner are given in this section.

## A.1   Rewards

The three reward types of the manipulation planner are visualized in Fig. 5.

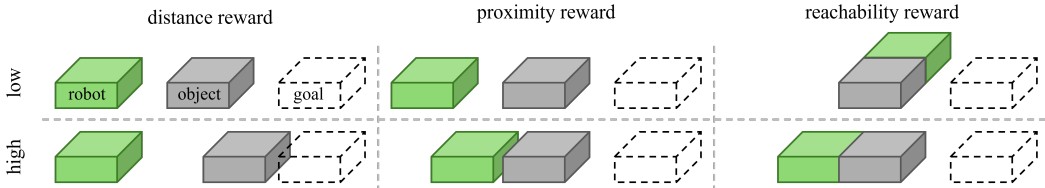

Figure 5: Low and high reward states for the three rewards of the manipulation planner.

**Reachability reward clipping**   The reachability measure $m$ defined in (8) is clipped from below to a lower bound value $m_{\min} \in \mathbb{R}_+$ in our implementation to prevent overly large rewards, i.e., $\overline{m} = \max(m, m_{\min})$, and the reward calculation (9) is updated to

$$r_{\mathrm{m}}(\overline{m}) = -q_{\mathrm{m}} \log\left(\frac{\overline{m}}{m_{\min}}\right). \tag{12}$$

## A.2   Node selection and extension horizon

**Pareto distribution**   The continuous truncated Pareto distribution is defined as

$$p(x; n_{\mathrm{n}}, \beta) = \frac{\beta x^{-(\beta+1)}}{1 - n_{\mathrm{n}}^{-\beta}}, \tag{13}$$

for any $x \in [1, n_{\mathrm{n}}]$, and $\beta$ determines the width of the distribution. The probability of sampling a specific node index $i_{\mathrm{n}} = i$ is

$$P(i_{\mathrm{n}} = i; n_{\mathrm{n}}, \beta) = \int_{i}^{i+1} p(x; n_{\mathrm{n}}, \beta)\,\mathrm{d}x = \frac{i^{-\beta} - (i+1)^{-\beta}}{1 - n_{\mathrm{n}}^{-\beta}}. \tag{14}$$

**Varying search parameters**   Instead of using fixed values for the Pareto distribution parameter $\beta$ (greediness of the node selection) and the extension horizon (related to the task complexity), these values are modified during the search. The values are updated depending on whether the search progresses or not. The greediness of the node selection should be reduced when stuck in a local minima, and the extension horizon can be interpreted as a measure of task complexity. An intuitive example of the meaning of the extension horizon is the Rubik's cube, where one must execute a sequence of actions to improve the cube's configuration. Here, some worse intermediate states are necessary to improve the overall configuration.

---

**Algorithm 2** Node Extension Parameter Update

1: $\cdots$
2: $i_{\mathrm{n}}, n_{\mathrm{e}} = \texttt{node\_selection}(\beta, n_{\mathrm{e}})$ ▷ Alg. 1, ln. 6
3: $\texttt{better\_node} = \texttt{false}$
4: **for** $i_{\mathrm{e}}$ in $1 : n_{\mathrm{e}}$ **do** ▷ Alg. 1, ln. 7
5:   $\boldsymbol{a}_{n_{\mathrm{n}}+1} = \texttt{action\_sampling}(p_{\mathrm{a}})$
6:   $\boldsymbol{s}_{n_{\mathrm{n}}+1} = \texttt{extension}(\boldsymbol{s}_{n_{\mathrm{n}}}, \boldsymbol{a}_{n_{\mathrm{n}}+1})$
7:   **if** $\texttt{is\_best\_node}(\boldsymbol{s}_{n_{\mathrm{n}}+1})$ **then**
8:     $\beta = \beta_{\max}$
9:     $n_{\mathrm{e}} = 0.95 n_{\mathrm{e}} + 0.05(i_{\mathrm{e}} + 1)$
10:     $\texttt{better\_node} = \texttt{true}$
11:   $n_{\mathrm{n}} = n_{\mathrm{n}} + 1$
12:   $i_{\mathrm{n}} = n_{\mathrm{n}}$
13: **if** $\texttt{better\_node} = \texttt{false}$ **then**
14:   $\beta = \max(0.99\beta, \beta_{\min})$
15:   $n_{\mathrm{e}} = \min(0.95 n_{\mathrm{e}} + 0.05(n_{\mathrm{e}} + 1), n_{\mathrm{e},\max})$
16:   $n_{\mathrm{n}} = n_{\mathrm{n}} + 1$ ▷ Alg. 1, ln. 10
17: $\cdots$

---

We bound $\beta \in [\beta_{\min}, \beta_{\max}]$ with lower and upper bound $\beta_{\min}, \beta_{\max} \in \mathbb{R}_+$, respectively, and we bound $n_{\mathrm{e}} \in [1, n_{\mathrm{e},\max}]$ with upper bound $n_{\mathrm{e},\max} \in \mathbb{N}_+$. In the inner-most for-loop of Alg. 1 (ln.

7), we check if the search progresses, i.e., if a new node with the overall highest reward is found. In this case, we update the parameters so the node selection is greedier and the extension horizon approaches the number of extensions required for search progress. If no new best node is found, we update the parameters so that the node selection is less greedy and the extension horizon approaches the upper horizon bound. These updates to Alg. 1 are stated in Alg. 2.

## A.3 Actions

The four action types of the manipulation planner are visualized in Fig. 6.

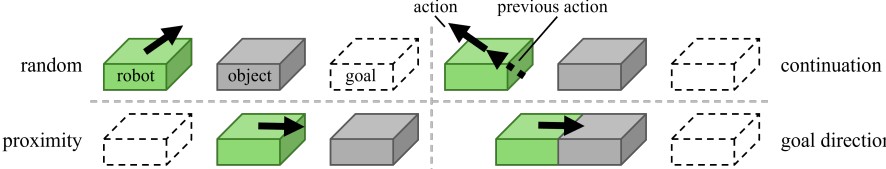

Figure 6: Visualization of the four action types of the manipulation planner.

**Varying action step size**    It can be helpful to have actions with different time step sizes to allow both short, precise motions and long, far-reaching motions. We randomly sample the action step size as an integer multiple of the base step size $\Delta t_\mathrm{a}$, i.e.,

$$\overline{\Delta t}_\mathrm{a} = k\Delta t_\mathrm{a}, \quad \text{with } k \sim \mathcal{U}(1, k_{\max}), \tag{15}$$

where $k_{\max} \in \mathbb{N}$ is the upper bound for the step size scaling. After the planner's search is finished, we split all actions to the base step size to obtain one common action step size for learning.

**Closed-form solution for goal-directed action**    The closed-form solution for the optimal control problem (11) is

$$\Delta \boldsymbol{a}_0 = -\left(\boldsymbol{B}^\mathsf{T}\boldsymbol{Q}\boldsymbol{B} + \boldsymbol{R}\right)^{-1}\left(\boldsymbol{B}^\mathsf{T}\boldsymbol{Q}\left(\boldsymbol{f}_0^* - \boldsymbol{s}_\mathrm{g}\right) + \boldsymbol{R}\boldsymbol{a}_0^*\right). \tag{16}$$

## A.4 Tree extension

**Dynamics rollout**    The actions of the planner are position commands tracked by a low-level PD controller $\boldsymbol{u}_t = -K_\mathrm{p}(\boldsymbol{q}_{\mathrm{r},t} - \boldsymbol{a}_t) - K_\mathrm{d}\dot{\boldsymbol{q}}_{\mathrm{r},t}$ with controller gains $K_\mathrm{p}$ and $K_\mathrm{d}$. A low-level control step size $\Delta t_\mathrm{c} \le \Delta t_\mathrm{a}$ smaller than the action step size is used to generate a position reference

$$\boldsymbol{a}_{\mathrm{ref},t} = \left(\boldsymbol{q}_{\mathrm{r},j_\mathrm{n}} + \boldsymbol{a}_{j_\mathrm{n}}\right)\frac{\Delta t_\mathrm{a} - t\Delta t_\mathrm{c}}{\Delta t} + \left(\boldsymbol{q}_{\mathrm{r},i_\mathrm{n}} + \boldsymbol{a}_{i_\mathrm{n}}\right)\frac{t\Delta t_\mathrm{c}}{\Delta t}. \tag{17}$$

The reference is a linear interpolation between the last commanded position for the parent node, $\boldsymbol{q}_{\mathrm{r},j_\mathrm{n}} + \boldsymbol{a}_{j_\mathrm{n}}$, and the last actual position plus the newly computed relative action, $\boldsymbol{q}_{\mathrm{r},i_\mathrm{n}} + \boldsymbol{a}_{i_\mathrm{n}}$, which creates a continuous control input. The same control scheme is used for RL policy actions.

## A.5 Parameters

The planner's parameters that are not task-specific are listed in Table 2.

| Parameter | Value | Parameter | Value |
|---|---|---|---|
| goal bias $b_\mathrm{g}$ | 0.0 | Pareto parameter upper bound $\beta_{\max}$ | 1.2 |
| number of goals $n_\mathrm{g}$ | 30 | Pareto parameter lower bound $\beta_{\min}$ | 0.2 |
| number of extensions $n_\mathrm{e}$ | 30 | extension horizon bound $n_{\mathrm{e},\max}$ | 10 |
| reachability reward bound $m_{\min}$ | 0.001 | base action time step $\Delta t_\mathrm{a}$ | 0.4s |
| action time step scale bound $k_{\max}$ | 3 | | |

Table 2: Planner parameters.

# B Reinforcement learning implementation details

Implementation details and parameters for reinforcement learning are given in this section.

## B.1 Pure DDPG

Our reinforcement learning setup largely follows the algorithm outlined in [53]. The policy network consists of four hidden layers with 256 neurons each. We use ReLU and tanh output activation functions to ensure the actions lie within [-1, 1]. The value function network also uses four hidden layers with 256 nodes each and ReLU activations but no output activation. We clip the value function's objective to the minimum value possible for the rollout horizon to avoid the unbounded divergence of value estimates. To improve stability during training, we employ target networks with Polyak averaging updates for both the policy and the value function. A normalizer estimates the mean and standard deviation for each input and normalizes all input values to zero mean and unit variance. The exploration noise consists of actions chosen uniformly at random with a probability

| Hyperparameter | Value |
|---|---|
| policy layers | 4 |
| value function layers | 4 |
| neurons per layer | 256 |
| policy learning rate | 0.001 |
| value function learning rate | 0.001 |
| Polyak averaging factor $\tau$ | 0.05 |
| random action chance $\eta$ | 0.3 |
| discount factor $\gamma$ | 0.98 |
| epochs $n_{\text{epochs}}$ | task specific |
| cycles $n_{\text{cycles}}$ | 50 |
| rollouts $n_{\text{rollouts}}$ | 2 |
| rollout horizon $n_{\text{steps}}$ | task specific |
| training episodes $n_{\text{episode}}$ | 40 |
| training batch size $n_{\text{batch}}$ | 256 |
| replay buffer size | $10000N$ |
| evaluation runs per epoch | 10 |

Table 3: Training hyperparameters.

of $\eta$ and additive Gaussian noise for the actions chosen by the policy. A complete list of hyperparameters is given in Table 3, and the whole algorithm is stated in Alg. 3. We stop the training if the policy achieves a success rate of 1.0 or if the maximum number of epochs has been reached.

---

**Algorithm 3** DDPG Algorithm

---
1: Initialize policy $\pi_{\boldsymbol{\theta}}$ and value function $Q$ with random weights
2: Initialize targets $Q'$ and $\pi'$ with weights $\phi' = \phi$, $\boldsymbol{\theta}' = \boldsymbol{\theta}$
3: Initialize replay buffer $\boldsymbol{B}$
4: **for** epoch in $1 : n_{\text{epochs}}$ **do**
5:   **for** cycle in $1 : n_{\text{cycles}}$ **do**
6:     **for** rollout in $1 : n_{\text{rollouts}}$ **do**
7:      Sample start state $\boldsymbol{s}_0$ and goal state $\boldsymbol{s}_{\text{g}}$
8:      **for** $t$ in $1 : n_{\text{steps}}$ **do**                      ▷ policy rollout
9:       **if** $\text{rand}() \leq \eta$ **then**
10:        Sample uniformly random action $\boldsymbol{a}_t$
11:       **else**
12:        Compute action $\boldsymbol{a}_t = \pi_{\boldsymbol{\theta}}(\boldsymbol{s}, \boldsymbol{s}_{\text{g}}) + \mathcal{N}_t$, with exploration noise $\mathcal{N}_t$
13:      Take action $\boldsymbol{a}_t$ and obtain new state $\boldsymbol{s}_{t+1}$
14:      Store transition $(\boldsymbol{s}_t, \boldsymbol{a}_t, \boldsymbol{s}_{\text{g},t}, r_t, \boldsymbol{s}_{t+1})$ in $\boldsymbol{B}$
15:     Update normalizer with new samples            ▷ normalizer update
16:     **for** episode in $1 : n_{\text{episode}}$ **do**           ▷ network training
17:      Sample minibatch of $n_{\text{batch}}$ transitions $(\boldsymbol{s}_t, \boldsymbol{a}_t, \boldsymbol{s}_{\text{g},t}, r_t, \boldsymbol{s}_{t+1})$ from $\boldsymbol{B}$
18:      Compute value target $v_i = r_i + \gamma Q'_{\phi'}(\boldsymbol{s}_{i+1}, \pi'_{\boldsymbol{\theta}'}(\boldsymbol{s}_{i+1}, \boldsymbol{s}_{\text{g},i+1}), \boldsymbol{s}_{\text{g},i+1})$
19:      Compute value function gradient $\boldsymbol{g}_Q$ for $l_Q = \frac{1}{n_{\text{batch}}} \sum_i (v_i - Q_{\boldsymbol{\theta}}(\boldsymbol{s}_i, \boldsymbol{a}_i, \boldsymbol{s}_{\text{g},i}))^2$
20:      Compute policy gradient $\boldsymbol{g}_{\pi}$ for $l_{\pi} = \frac{1}{n_{\text{batch}}} \sum_i Q_{\boldsymbol{\theta}}(\boldsymbol{s}_i, \pi_{\phi}(\boldsymbol{s}_i, \boldsymbol{s}_{\text{g},i}), \boldsymbol{s}_{\text{g},i})$
21:      Apply the update to $\pi_{\phi}$ and $Q_{\boldsymbol{\theta}}$
22:     Update the target networks: $\phi' = \tau\phi + (1-\tau)\phi'$, $\boldsymbol{\theta}' = \tau\boldsymbol{\theta} + (1-\tau)\boldsymbol{\theta}'$
23:   Evaluate policy and stop if success rate $= 1.0$

---

## B.2 Demonstrations in replay buffer

Instead of generating all data with an online policy rollout, data from the planner's search tree can be used with a bias $b_p \in [0,1]$. Accordingly, the rollout loop of Alg. 3 (ln. 6) is modified to use both the policy and the planner to generate trajectories, as stated in Alg. 4.

Planner demonstration lengths vary and can be shorter or longer than the rollout horizon $n_{\mathrm{steps}}$. For long trajectories, the beginning is cut so that only the last $n_{\mathrm{steps}}$ transitions remain since these transitions are most likely to contain reward signals.

---

**Algorithm 4** Planner Rollout

1: $\cdots$
2: **for** rollout in $1 : n_{\mathrm{rollouts}}$ **do**  $\triangleright$ Alg. 3, ln. 6
3:   **if** $\mathrm{rand}() \le b_p$ **then**
4:     Sample goal state $s_g$
5:     Find node closest to $s_g$
6:     Find path from root node to closest node
7:     Clip or pad trajectory if necessary
8:     Store all transitions in $B$
9:   **else**
10:     Sample $s_0$ and $s_g$  $\triangleright$ Alg. 3, ln. 7
11:     **for** $t$ in $1 : N$ **do**  $\triangleright$ Alg. 3, ln. 8
12: $\cdots$

---

Short trajectories are padded by repeating the planner's start state to reach a length of $n_{\mathrm{steps}}$. The action commands for the padded start states are the initial joint states of the robot.

## B.3 Pre-training

A policy network $\pi_{\boldsymbol{\theta}_{\mathrm{IL}}}$ of the same size as $\pi_{\boldsymbol{\theta}}$ can be trained with imitation learning from planner demonstrations. The policy is trained with input $(s, s_g)$ and output $(a)$ from the demonstrations. Similarly, the value function is trained with input $(s, s_g, a)$ and output $(v)$, where the discounted value $v$ for an input tuple can be calculated trivially from the rewards (1) for all states in a demonstration trajectory. Algorithm 5 states the pre-training procedure. Note that the policy and the value function can be pre-trained separately.

The pre-trained policy and value function are used as described in [25]. In summary, during the policy rollout and network training phase, the better one of the two policies, as evaluated by the value function, is used. Note that we use the same value function for pre-training and online training.

---

**Algorithm 5** Pre-training

1: **for** demo in $n_{\mathrm{demos}}$ **do**  $\triangleright$ demo generation
2:   Sample goal state $s_g$
3:   Find node closest to $s_g$
4:   Find path from root node to closest node
5:   Store all transitions in $B$ if goal reached
6:   Compute value for all transition
7: **for** episode in $1 : n_{\mathrm{episode}}$ **do**  $\triangleright$ network training
8:   Sample minibatch of $n_{\mathrm{batch}}$ tuples $(s_t, a_t, s_{g,t}, v_t)$ from $B$
9:   Compute value function gradient $g_Q$ for $l_Q = \frac{1}{n_{\mathrm{batch}}} \sum_i (v_i - Q_{\boldsymbol{\theta}}(s_i, a_i, s_{g,i}))^2$
10:   Compute policy gradient $g_\pi$ for $l_\pi = \frac{1}{n_{\mathrm{batch}}} \sum_i \|a_i - \pi_\phi(s_i, s_{g,i})\|^2$
11:   Apply the update to $\pi_\phi$ and $Q_{\boldsymbol{\theta}}$

---

# C Experiment details

The evaluation systems are described in this section. Afterward, several ablations and comparisons for the manipulation planner and reinforcement learning are provided.

## C.1 Evaluation systems

Table 4 provides the task-specific parameters for the evaluation systems. Note that all object velocity states have a scaling $Q_{\mathrm{d,o},q} = 0.1 I$ and the robot state scaling is zero: $Q_{\mathrm{d,r}} = 0 I$. A description of each system is provided below.

**Box push** The box push task is a one-dimensional pushing task where one actuated pusher box (robot) must push a second unactuated box (object) to a goal position on a line. Due to its simplicity, this task can give valuable insight into the general properties of planning and learning. Moreover, only pushing but no pulling is possible, which creates an interesting manipulation challenge.

| System | $n_{\text{epochs}}$ | $n_{\text{steps}}$ | $p_a$ | $\boldsymbol{Q}_{\text{d,o},\boldsymbol{q}}$ | $\boldsymbol{Q}_{\text{p}}$ | $q_{\text{m}}$ |
|---|---|---|---|---|---|---|
| Box push | 150 | 75 | [1 1 2 2] | $\text{diag}(1)$ | $\boldsymbol{I}$ | 1 |
| Box push 2D | 150 | 100 | [6 2 2 1] | $\text{diag}(1, 1)$ | $0.001\boldsymbol{I}$ | 0.001 |
| Planar hand | 150 | 80 | [1 1 2 2] | $\text{diag}(1, 1, \pi/2)$ | $0.01\boldsymbol{I}$ | 0.01 |
| Floating hand | 150 | 100 | [1 1 1 1] | $\text{diag}(1, 2, \pi/2)$ | $0.01\boldsymbol{I}$ | 0.01 |
| Allegro hand | 300 | 50 | [1 3 2 2] | $\text{diag}(100, 100, 100, 1, 1, 10)$ | $0.01\boldsymbol{I}$ | 0.01 |
| Box grasp | 300 | 50 | [0 1 1 1] | $\text{diag}(1, 1, 10, 0, 0, 0)$ | $0.01\boldsymbol{I}$ | 0.01 |
| Ball turn | 300 | 75 | [2 1 3 3] | $\text{diag}(1, 1, 1, 2, 2, 0)$ | $0.1\boldsymbol{I}$ | 0.1 |
| Stool lift | 300 | 75 | [1 1 1 1] | $\text{diag}(1, 1, 1, 2, 2, 0)$ | $\boldsymbol{I}$ | 0.1 |

Table 4: Task parameters.

**Box push 2D**   The two-dimensional box push task is the direct extension of the one-dimensional box push task. Just as the one-dimensional box push task, it is an illustrative example. The complexity of the task is increased by placing the pusher box between the object box and the goal. This setup requires the pusher to move around the obstacle before pushing it towards the goal.

**Planar hand**   The planar hand is a simplified representation of in-hand manipulation in a two-dimensional plane. A hand with two two-link fingers is used to rotate and lift a box. Even with the restriction to a plane, this task requires dexterous manipulation and gives insight into the planner's ability to solve in-hand manipulation tasks.

**Floating hand**   The floating hand has the same kinematic as the planar hand and is also limited to a two-dimensional plane. However, instead of a fixed wrist, the floating hand can move in the plane. The task is to reorient a box and move it in space. This task combines dexterous manipulation as well as picking and placing an object, two commonly required skills.

**Allegro hand**   The Allegro hand is a robotic hand by Wonik Robotics with four fingers, sixteen actuated finger joints, and two actuated wrist joints. The task is to reorient a cube by turning it around the upright z-axis. In-hand manipulation is a commonly investigated robotic skill due to its high dimensionality and required dexterity.

**Box grasp**   The box grasp task uses a stationary quadrupedal robot with a mounted arm to grasp and lift a box with a handle. It is a classic pick-and-place task, where finding a proper grasp is the main challenge. This task demonstrates our method's ability to solve pick-and-place scenarios on real systems.

**Ball turn**   The ball turn task involves a quadrupedal robot on its back that is using its legs to reorient a ball. This setup resembles in-hand manipulation tasks where multiple multi-joint fingers (legs) are used to reorient an object. We use this setup to demonstrate our method's ability to perform in-hand manipulation on real systems.

**Stool lift**   The stool lift task comprises two stationary quadrupedal robots with mounted arms that lift a bar stool into an upright configuration. The robots are deliberately prevented from grasping the stool by fixing their grippers in a closed state. In this manner, whole-body and bimanual manipulation of the stool is required, with contact occurring on multiple parts of the robot arms. We use this task as an exemplary real-world cleaning scenario, where robots can help put moved or fallen furniture back into desired configurations. By forcing whole-body contact, we also demonstrate our method's ability to perform whole-body manipulation on real systems.

## C.2   Manipulation Planner

We present additional search results for two tasks and several studies on the planner's components and parameters.

### C.2.1 Additional tasks

**Box push and box push 2D results** Extending the results presented in Fig. 2, we show the search progress for the two additional tasks, box push and box push in 2D. The results are in line with the reported results for the other tasks.

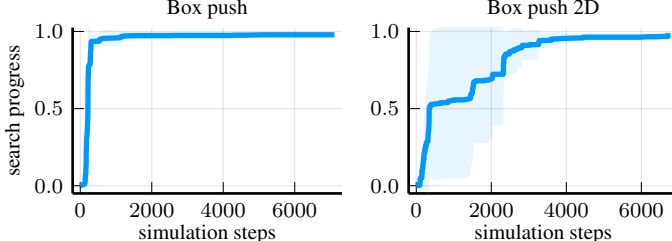

Figure 7: The performance of the Manipulation Planner on two additional tasks. Relative closeness to the goal for the best node in the tree so far vs. the total number of nodes (simulation steps).

### C.2.2 Baseline comparison

A comparison of three manipulation tasks with increasing difficulty based on the planar hand is given in Fig. 8. The tasks are (1) in-hand reorientation of a box by 90°, in-hand reorientation of a box by 180°, and throwing the box onto a shelf. The 90° turn can be achieved by simply tipping over the box. For 180°, regrasping is required. Throwing the box onto the shelf requires highly dynamic movements to overcome the gap between the hand and the shelf. We compare our planner to the quasi-static planner in [14].

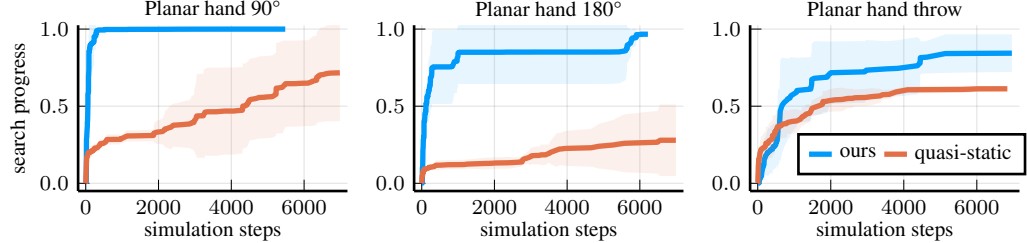

Figure 8: Performance comparison of our planner with the state-of-the-art planner introduced in [14] for using the planar hand to reorient a box 90°, 180°, and throw the box into a desired location. Relative closeness to the goal for the best node in the tree so far vs. the total number of nodes (simulation steps).

We outperform the quasi-static planner across all three tasks. It is important to note that, for the throwing task, the search progress does not fully capture performance. While our planner consistently succeeds in throwing the box onto the shelf, the quasi-static planner fails to find this solution. These results underscore the necessity of dynamic planning for advanced manipulation tasks and highlight our planner's superior capability to solve such tasks, thereby outperforming a state-of-the-art planner.

### C.2.3 Ablations

**Extension horizon** The extension horizon specifies how many sequential actions are taken once a node has been selected to extend the tree. We assume that tasks of different complexity require different extension horizons. We evaluate different but fixed extension horizons for all tasks to measure the influence on the planner's search progress in Fig. 9. We also evaluate the automatically varying horizon that we use in our implementation as described in Appendix A.2. We measure the average closeness to the goal over the full search duration, which is larger for quicker search progress and higher final closeness. While a longer extension horizon tends to improve performance

overall, there is no strong correlation. At the same time, the varying horizon performs well in all cases with reduced variance compared to the fixed horizons.

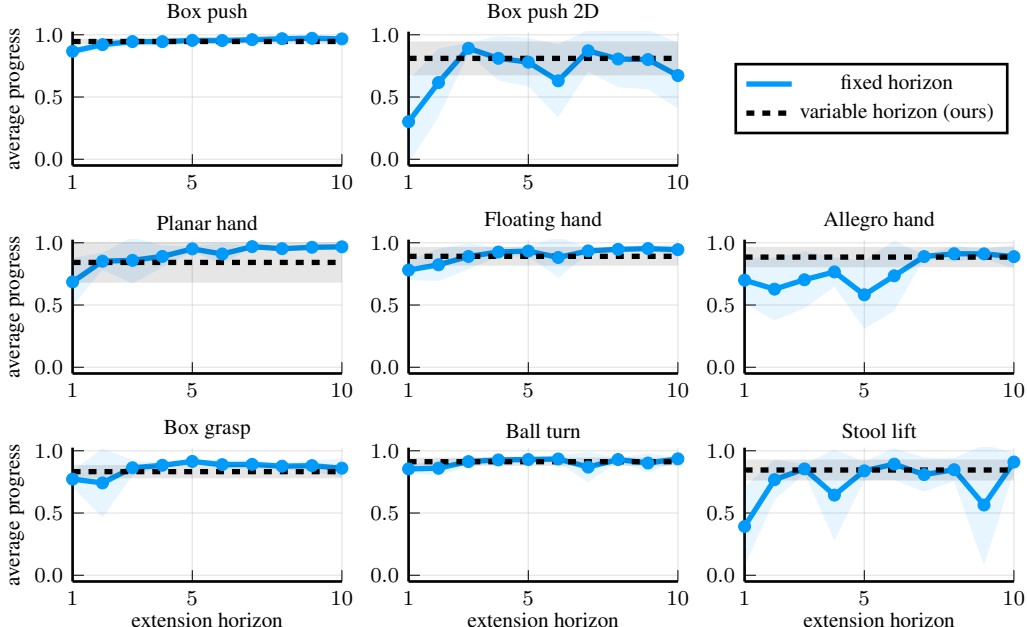

Figure 9: The performance of the Manipulation Planner for different extension horizons measured as the average closeness to the goal during the search. Comparison of different fixed horizons (blue) and variable horizon (black).

**Pareto distribution exponent** The Pareto distribution parameter $\beta$ modifies how greedily nodes with high rewards are extended. We evaluate different but fixed Pareto parameters for all tasks to investigate how the parameter influences the planner's search progress in Fig. 10. We also evaluate

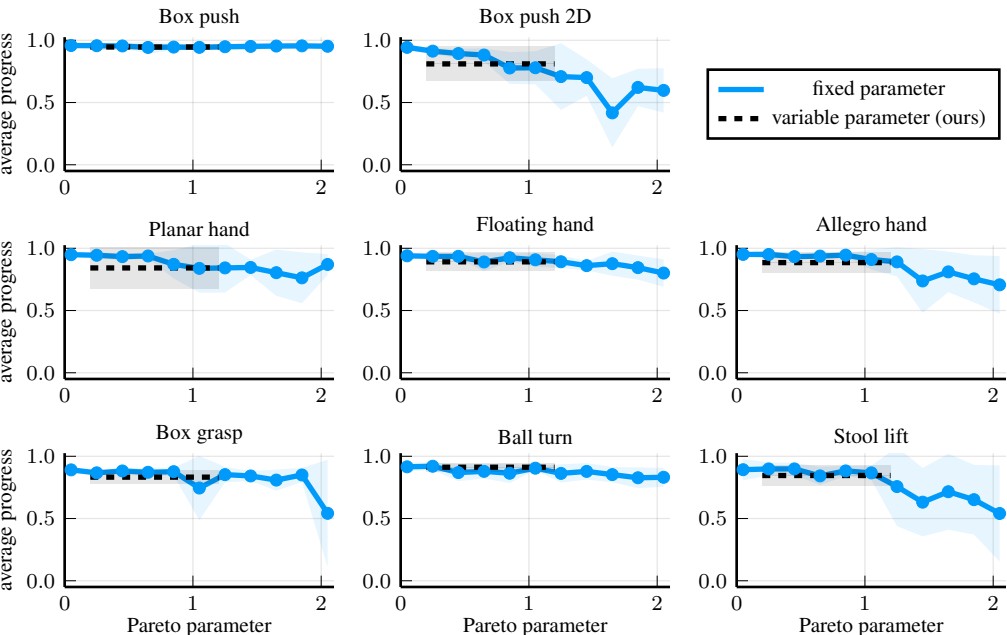

Figure 10: The performance of the Manipulation Planner for different Pareto parameters measured as the average closeness to the goal during the search. Comparison of different fixed parameters (blue) and variable parameter (black).

the automatically varying parameter that we use in our implementation as described in Appendix A.2. For intuition, a parameter $\beta = 0.05$ means that the currently best node will be selected with a probability of roughly $P = 10\%$, whereas $\beta = 2.0$ means a probability of $P = 75\%$. There is a consistent trend that the planner's performance decreases as the node selection becomes more greedy (the parameter increases). This effect could be caused by the node selection becoming so greedy that the search gets stuck in local minima by repeatedly selecting the same few nodes. Using an automatically varying range for the parameter does not lead to an improvement over always choosing a low parameter. Apparently, the Pareto distribution with a low $\beta$ already has a good tradeoff between exploitation and exploration.

**Sub-goal ratio**   To assess the usefulness of having sub-goals during the planner's search, as in rapidly exploring random trees, we evaluate different ratios of sub-goals to node extensions in Fig. 11. A goals/extensions ratio of $1/900$ means we choose one goal and do 900 extensions for this single goal, whereas a ratio of $900/1$ means we choose 900 goals and do one extension for each of these goals. Note that in this evaluation, we always do a total of 900 extensions. There is no consistent trend indicating that more or less sub-goals improve search performance. Multiple explanations exist for these results: Our tasks could have very few local minima. However, since the Pareto parameter does influence search performance (cf. Fig. 10), there may be many local minima, but our node selection strategy is already good enough to avoid these. Yet another possibility is that the difference in sub-goals is too small to provide a benefit over searching with just a single goal.

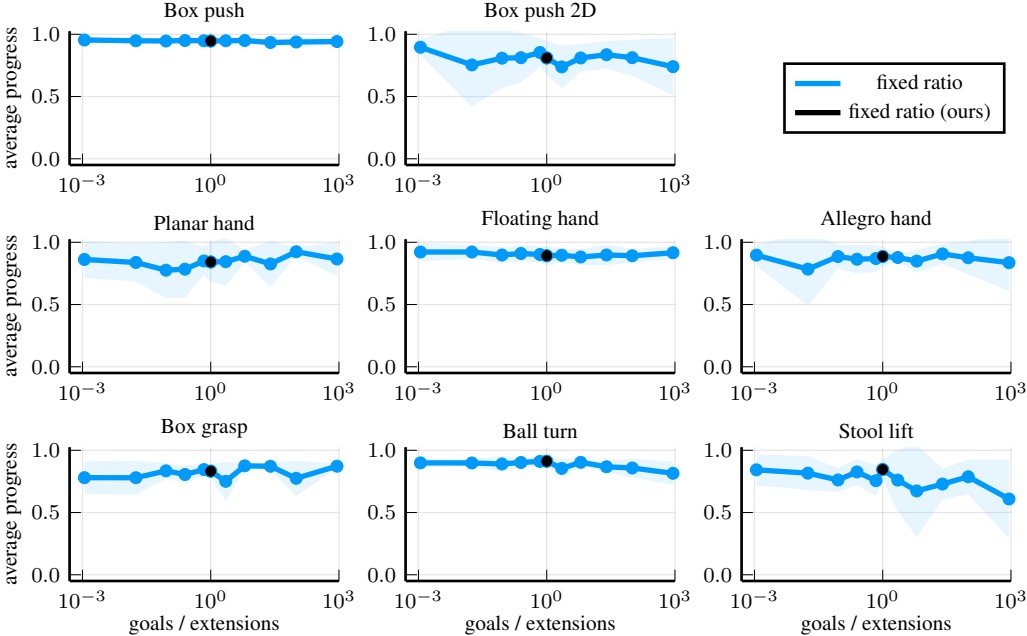

Figure 11: The performance of the Manipulation Planner for different ratios of goals/extensions measured as the average closeness to the goal during the search.

**Action types**   We investigate the effectiveness of different combinations of action types with just random actions as a baseline in Fig. 12. The fact that purely random actions already lead to good results in many tasks indicates that the other components of the planner, such as the heuristic node selection and specific rewards, lead to meaningful search progress. Still, proximity and goal-directed actions alongside random actions in the search further improve the performance for some tasks.

**Proximity and reachability reward scaling**   The effect of different proximity and reachability reward scalings is shown in Fig. 13. For most tasks, the search fails once the proximity reward becomes too high, potentially because the necessary distance between the robot and the object for

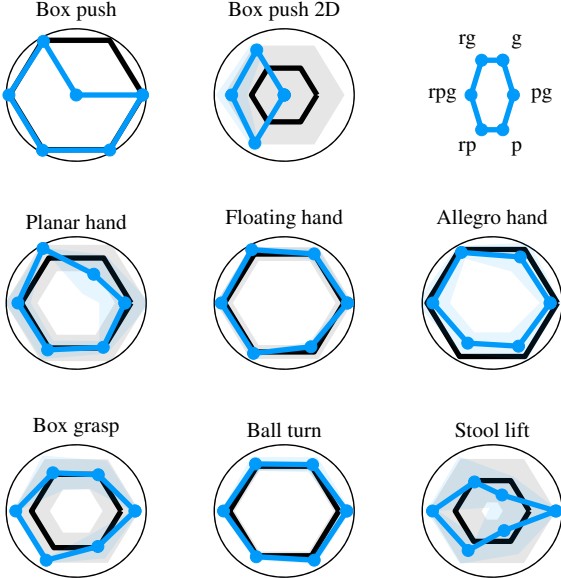

Figure 12: The performance of the Manipulation Planner for different combinations of action types measured as the average closeness to the goal during the search. Comparison of the following action types (blue): goal-directed (g), random+goal-directed (rg), random+proximity+goal-directed (rpg), random+proximity (rp), proximity (p), proximity+goal-directed (pg), and random (black).

reconfiguration is punished too severely. The reachability reward scaling shows a similar, although weaker, trend. Especially for the more complex tasks involving the Allegro hand or robot arms, it is difficult to clearly state how to scale the rewards, and a parameter sweep is necessary to find the optimal value. It can be assumed that the same effect would occur for dense-reward reinforcement learning, highlighting one of the benefits of the planner, as parameter sweeps can be executed much faster with the planner than with learning.

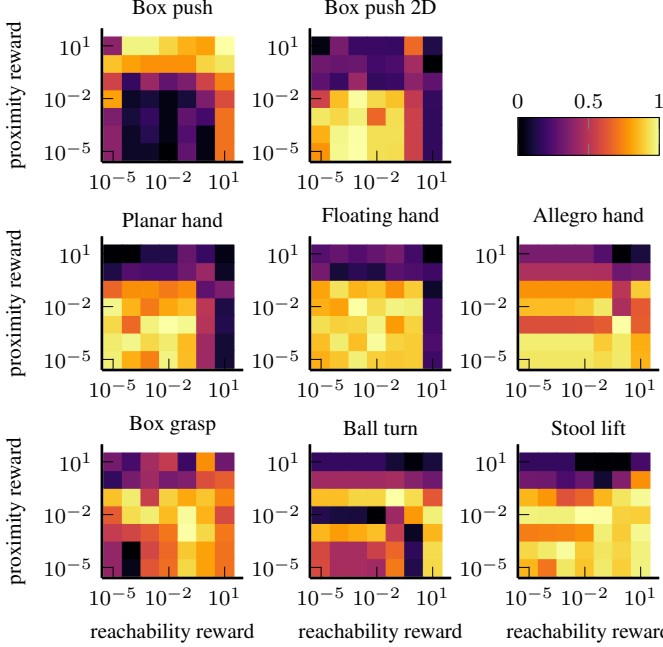

Figure 13: The performance of the Manipulation Planner for different combinations of action types measured as the average closeness to the goal during the search. Comparison of different proximity and reachability reward scalings. Purple is bad, yellow is good.

## C.3 Reinforcement learning

We present additional learning results for two tasks and several comparisons for different learning methods and demonstration usage.

### C.3.1 Additional tasks

**Box push and box push 2D results**   Extending the results presented in Fig. 3, we show the training progress for the two additional tasks, box push and box push in 2D. The results are in line with the reported results for the other tasks.

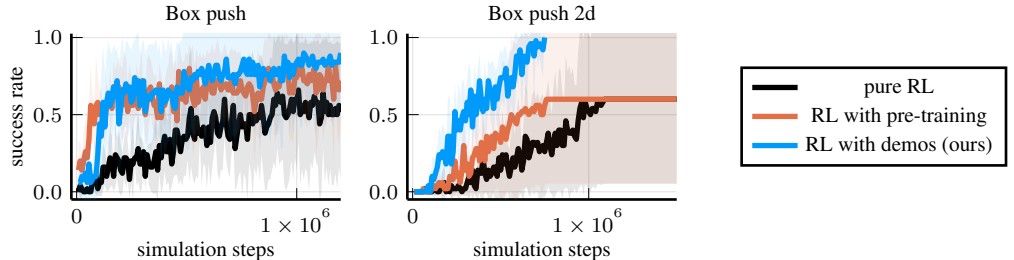

Figure 14: The performance of reinforcement learning for manipulation on two additional tasks. Comparison of our approach of adding demonstrations to the replay buffer (blue), RL with an additional imitation policy pre-training (red), and pure RL (black).

### C.3.2 Planner and policy comparison

While the planner is able to solve the manipulation tasks presented in this paper, distilling a policy from planner demonstrations leads to more optimal and faster solutions (after training).

**Solution length comparison**   The trajectory lengths (in steps) for planning and learning are given in Tab. 5. We list the average trajectory lengths for five seeds.

The results show that both the planner and the trained policies are able to find solutions. However, the policies always require significantly fewer steps to solve the tasks, indicating a more optimized performance.

| Task | Planner | Policy |
|---|---|---|
| Box push | 80.4 | 59.4 |
| Box push 2D | 110.6 | 37.6 |
| Planar hand | 72.6 | 17.0 |
| Floating hand | 81.2 | 47.8 |
| Allegro hand | 41.0 | 30.4 |
| Box grasp | 43.4 | 21.2 |
| Ball turn | 34.0 | 21.2 |
| Stool lift | 57.8 | 28.4 |

Table 5: Solution trajectory lengths (in steps) of planner and trained policies for different manipulation tasks.

**Timing comparison**   The wall-clock times for planning and learning, as well as dimensions for the manipulation tasks, are given in Tab. 6. We list the average time for five seeds. *Single sol.* refers to finding a single trajectory leading to one goal. *Multiple sol.* refers to finding multiple solutions leading to different goal locations in order to generate a set of demonstrations for policy learning. *Policy training* refers to training the policy where demonstrations are fed to the replay buffer. The planner is no longer used at this stage. *Observation dim.* refers to the combined dimensions of state and goal space. *Action dim.* refers to the dimension of the action space.

The results show that both planning and policy learning scales to high-dimensional tasks with only a moderate increase in computation time.

### C.3.3 Baseline comparison

We provide a comparison to the state-of-the-art manipulation learning algorithm MoPA-RL [37] for the same three planar hand tasks described in Appendix C.2.2 and for a complex assembly task.

| Task | Single sol. | Multiple sol. | Policy training | Observation dim. | Action dim. |
|---|---|---|---|---|---|
| Box push | 3.1s | 15.3s | 19.6min | 8 | 1 |
| Box push 2D | 7.6s | 13.9s | 19.5min | 16 | 2 |
| Planar hand | 6.7s | 15.9s | 5.9min | 28 | 4 |
| Floating hand | 8.6s | 17.5s | 16.4min | 36 | 6 |
| Allegro hand | 3.8s | 43.7s | 67.0min | 98 | 18 |
| Box grasp | 7.7s | 40.0s | 68.9min | 54 | 7 |
| Ball turn | 4.1s | 57.6s | 58.1min | 74 | 12 |
| Stool lift | 8.5s | 56.2s | 46.4min | 84 | 14 |

Table 6: Wall clock times and dimensions for planning and policy learning of different manipulation tasks.

**Planar hand** We use sparse rewards for the baseline comparison on the planar-hand tasks and show the results in Fig. 15.

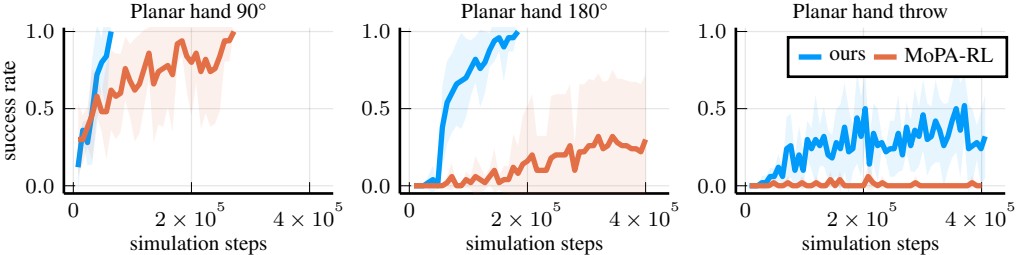

Figure 15: Performance comparison of our full algorithm with MoPA-RL [37] for using the planar hand to turn a box 90°, 180°, and throw the box into a desired location.

We outperform MoPA-RL for all three tasks. While the MoPA-RL is also able to quickly and consistently find solutions to the 90°-reorientation task, it has more difficulty with the 180° reorientation. Finally, MoPA-RL is not able to find any solution to the throwing task, whereas our method achieves a success rate of more than 50%. As before, the results demonstrate the need for dynamic planning and our planner's ability to solve highly dynamic tasks.

**Sawyer assembly** With the *Sawyer assembly* task described in [37], we demonstrate our ability to solve complex assembly tasks that require precise manipulation in cluttered environments. The task is challenging because colliding with the movable table prevents assembly. Accordingly, collisions must be avoided. We evaluate five seeds for our method, and we reuse the results shown in [37] for MoPA-RL.

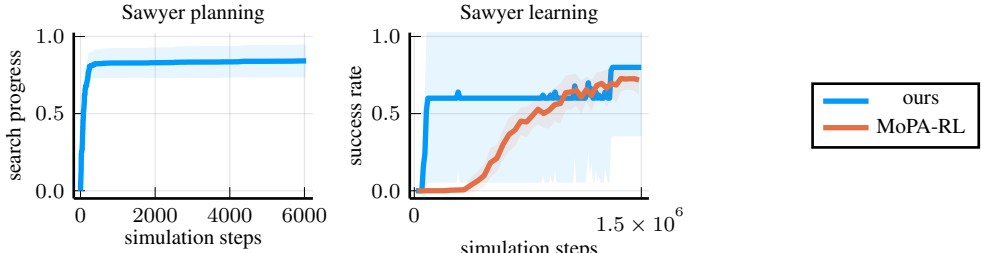

Figure 16: **Left**: The performance of our planner for solving the *Sawyer assembly* task presented in [37]. Relative closeness to the goal for the best node in the tree so far vs. the total number of nodes (simulation steps). **Right**: Performance comparison of our full algorithm with MoPA-RL [37] for the *Sawyer assembly* task. Results for MoPA-RL are taken from [37].

In the majority of cases, our planner is able to find solutions for this assembly task. Policy learning is successful in these cases with good planner demonstrations. While we achieve a similar average performance for policy learning as MoPA-RL, we have a higher variance due to the reliance on the planner's demonstrations. While our planner is designed for contact-rich manipulation, it can also

solve this assembly task where contacts must be avoided. Our method implicitly takes collision avoidance into account and matches the performance of MoPA-RL.

### C.3.4 Ablations

**Demonstration ratio** Adding demonstrations to the RL replay buffer improves performance compared to pure online learning, but pure offline learning with the demonstration data does not yield good results. Accordingly, there must be a tradeoff between online exploration and offline data. We investigate this tradeoff by varying the ratio of demonstrations in the replay buffer in Fig. 17. We measure the average reward over the entire training duration, which is larger for quicker learning progress and a higher final success rate. It appears that a small amount of demonstrations gives enough reward signals to bootstrap RL and enable exploration from a sub-optimal solution. Already having a low ratio of roughly $10\%$ yields satisfactory results, and we use $25\%$ as it gives the most robust performance for all tasks.

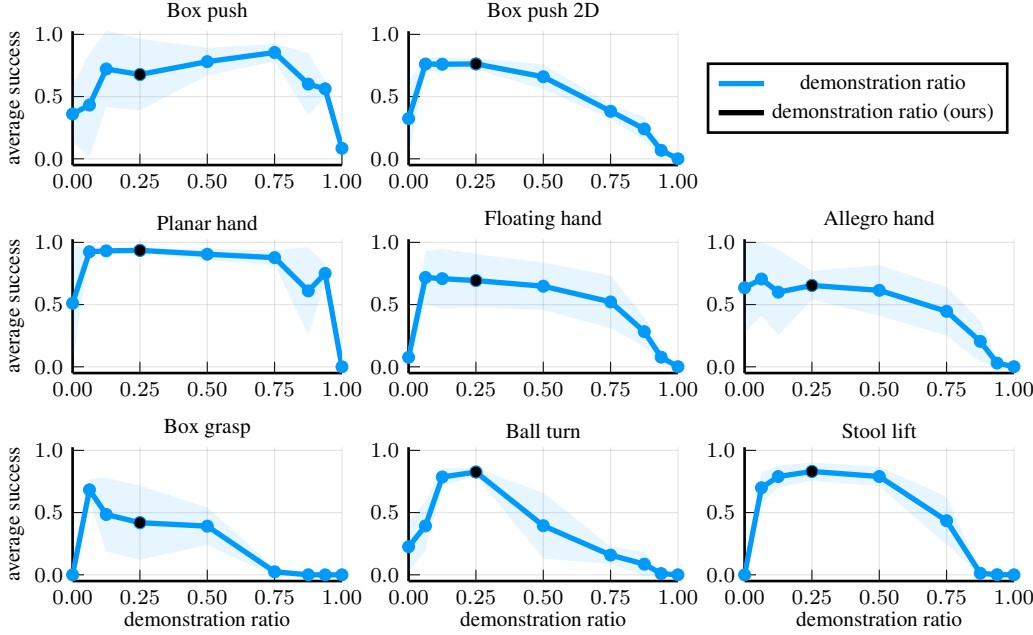

Figure 17: The performance of reinforcement learning (RL) for manipulation measured as the average success during training on eight tasks. Comparison of different demonstration-to-online-exploration ratios in the replay buffer. A ratio of 0.0 is pure online RL, 1.0 is pure offline RL.

**Demonstration inclusion comparison** Three different methods for adding demonstrations to the replay buffer are investigated. (1) Use a fixed ratio of demonstrations to online exploration. (2) Use a variable ratio, where the ratio decreases as the learning success rate increases. (3) Add a fixed number of demonstrations from the planner's tree only initially. As can be seen in Fig. 18, the specific method of introducing demonstrations into the replay buffer does not make a large difference. However, just adding demonstrations in the beginning is sometimes not enough to bootstrap learning successfully.

**Pre-training comparison** Directly using the pre-trained policy only works for the most simple box push task with a success rate of $0.10 \pm 0.13$. For all other tasks, the success rate was zero. We evaluated ten runs for each task. Apparently, the data generated by the planner is not good enough to directly perform imitation learning with it, and online exploration with RL is required to find successful policies.

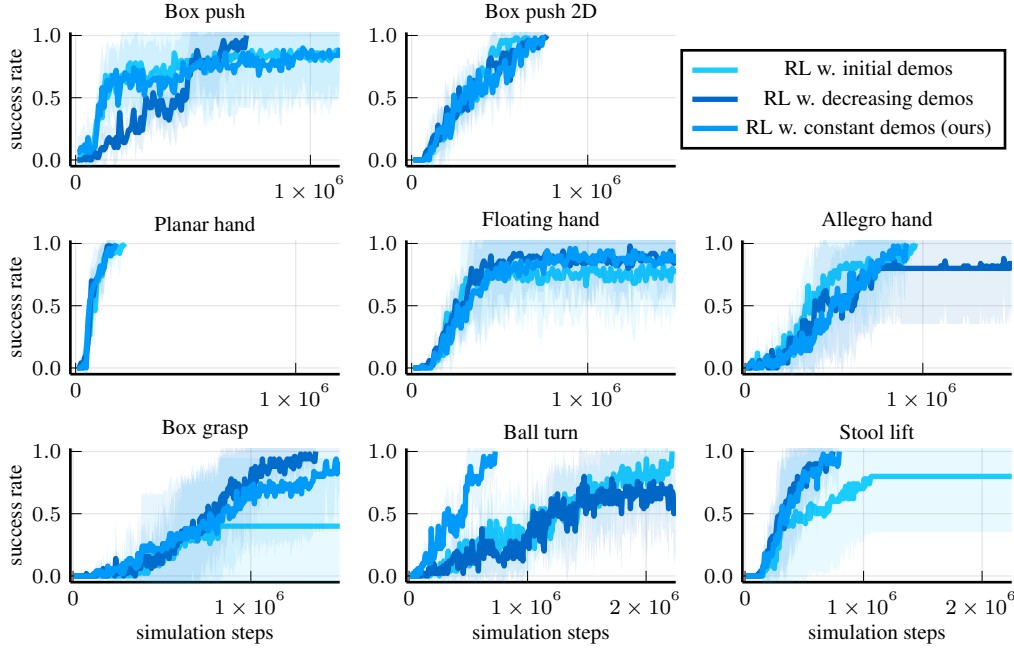

Figure 18: The performance of reinforcement learning for manipulation with demonstrations on eight tasks. Comparison of adding a constant ratio of demonstrations to the replay buffer (blue), adding fewer demonstrations with increasing success rate (dark blue), and adding demonstrations only at the beginning (purple).

Accordingly, we compare three different pre-training settings in Fig. 19. (1) Only pre-training a policy $\pi_{IL}$ and using it alongside the online RL policy. (2) Pre-training a policy and the value function. (3) Pre-training the policy and value function as well as adding a fixed ratio of demonstrations to the replay buffer. While adding a pre-trained policy to RL improves performance over pure DDPG, it does not significantly and consistently perform much better. Only once demonstrations are added to the replay buffer does the performance with a pre-trained policy improve significantly. Since the performance of adding demonstrations alongside pre-training is not better than just adding demonstrations, we attribute this improvement to using demonstrations and not to the pre-trained policy.

**HER comparison** For reference, we provide a comparison to hindsight experience replay (HER) in RL since this method can improve learning performance in sparse-reward settings. The results are shown in Fig. 20. While using HER improves the learning performance for some tasks compared to pure DDPG, it does not lead to any improvement for others and almost always performs worse than adding demonstrations to the replay buffer.

## C.4    Hardware evaluation

**Setup**    We use the quadrupedal Spot robot by Boston Dynamics with an arm arm mounted for certain tasks. Control and communication with the robot are achieved via the `bosdyn` software development kit. Accurate state estimation is achieved by using the OptiTrack motion capture system and placing markers on the objects. Communication between the Optitrack System and the robots was handled via ROS2, while gRPC was used to command policy actions with the Spot robots.

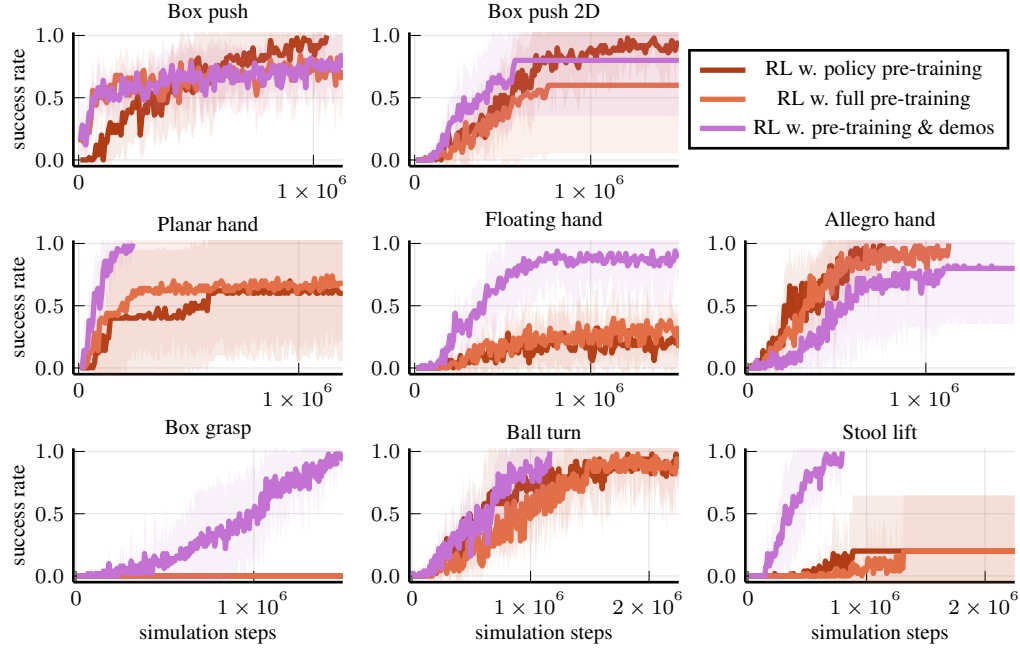

Figure 19: The performance of reinforcement learning for manipulation with pre-training on eight tasks. Comparison of pre-training just the policy (light red), pre-training the policy and value function (red), and pre-training the policy and value function as well as adding demonstrations (dark red).

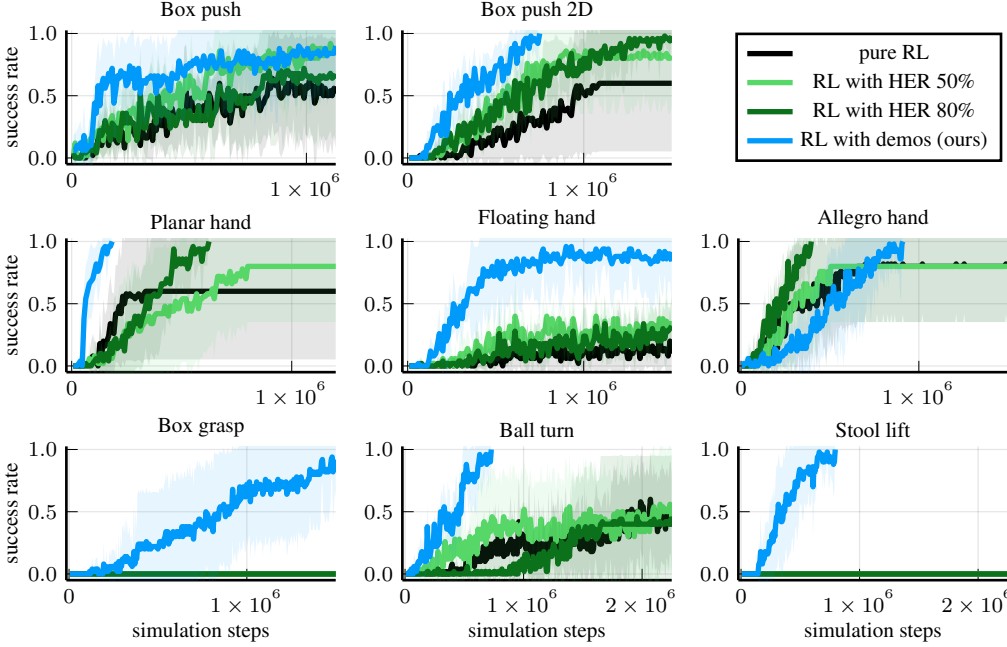

Figure 20: The performance of reinforcement learning for manipulation with hindsight experience replay on eight tasks. Comparison of our approach of adding demonstrations to the replay buffer (blue), RL with 50% HER relabeling (light green), RL with 80% HER relabeling (dark green), and pure RL (black).

**Box grasp** The box grasp task evaluates if the robot is able to grasp and lift the box. A trial is considered successful if the box is lifted off the ground. We evaluate 27 different starting locations by building a grid of start positions and orientations for the box:

$$x \in \{-0.1, 0.0, 0.1\} \times y \in \{-0.25, 0.00, 0.25\} \times \theta_z \in \{-45°, 0°, 45°\}. \tag{18}$$

Fig. 21 shows the progression of the box-grasp task. The challenge in this task is to learn that in order to lift the box, the handle must be grasped. Our trained policy is able to consistently grasp the handle and performs well.

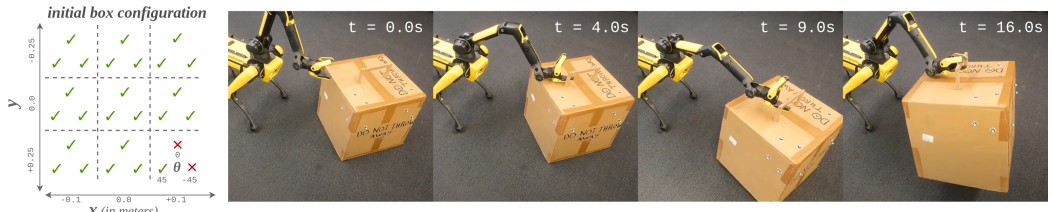

Figure 21: **Left**: the successful and failed initial configuration for the box-grasp tasks. **Right**: deployment of the trained policy for one Spot grasping and lifting the box.

**Ball turn** The ball-turn task is used as an in-hand-manipulation-equivalent example. A trial is considered successful if the box is rotated such that the angle deviation from the upright z-axis is below $40°$. We evaluate 20 different starting locations by testing a sequence of start rotations around the y-axis for the ball with increments of $15°$:

$$\theta_y \in \{\pm 180°, \pm 165°, \cdots, \pm 60°, \pm 45°\}. \tag{19}$$

Fig. 22 shows the progression of the ball-turn task. The robot can successfully perform some of the desired orientations. We attribute failures to the sim2real gap, with a rigid ball in simulation training and a soft yoga ball in the real-system setting.

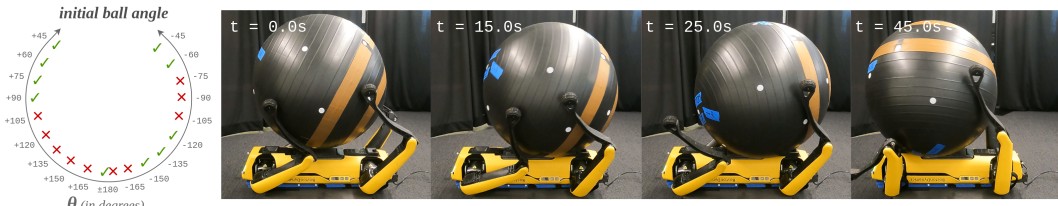

Figure 22: **Left**: the successful and failed initial orientations for the ball-turn tasks. **Right**: deployment of the trained policy for one Spot turning the ball to the upright position.

**Stool lift** The stool-lift task requires whole-body manipulation and is a bimanual task representative of a real-world cleaning task. A trial is considered successful if the robots maneuver the stool in the upright position. We evaluate 25 different starting locations by building a grid of start positions for the stool:

$$x \in \{-0.2, -0.1, 0.0, 0.1, 0.2\} \times y \in \{-0.2, -0.1, 0.0, 0.1, 0.2\}. \tag{20}$$

Fig. 4 shows the progression of the stool-lift task. Despite the challenging whole-body and bimanual setup, the robots frequently manage to lift the stool in the upright configuration. Failures can be attributed to a lack of measuring the robots' body positions. When contact forces occur, the robots' bodies are displaced (which we do not measure), leading to position offsets for the robot arms. In such cases, the policy has false state measurements and produces unsuccessful actions.

