# OpenReview forum: "Jacta: A Versatile Planner for Learning Dexterous and Whole-body Manipulation"
_robot-learning.org/CoRL/2024/Conference — CoRL 2024_

### Official Review · Reviewer_RBak · 2024-07-21
**The paper proposes to use motion-planning for bootstrapping several dextrous and whole-body manipulation tasks**

**Originality:** 3
**Technical Quality:** 4
**Clarity Of Presentation:** 4
**Potential Impact:** 3
**Recommendation:** 3
**Confidence:** 5

**Review:**

The paper proposes an interesting method for bootstrapping learning for complex manipulation tasks using motion planners as experts providing demonstrations. The paper is clearly written and well-organized. The paper also provides several simulation and real-world experiments, showcasing the efficacy and practicality of the proposed approach. While the results are mostly impressive, it is unclear if using motion planning techniques is scalable as well as generalizable to a lot of tasks, due to its non-real time computational delays as well as the dependency on access to the state space. Moreover, the final policy still relies on state information which is not readily accessible in the real-world, especially when deployed in-the-wild. Moreover, the requirement to predesign rewards for each of the state spaces makes it difficult to scale to a lot of manipulation problems with stochastic environments or where motion-planning typically fails such as where complex assembly or precise manipulation is required.

**Quality Of The Limitations Section:**

3

**Questions For Rebuttal:**

I request the authors to address the concerns below:
1. As acknowledged by the authors, the dependency of both the planner as well as the policy on the state space creates a bottleneck for this approach to scale to diverse tasks in the real-world. Another related work [1] further distills the state-based policy into an asymmetric policy which still relies on the state space for the critic, but distills the actor into a vision-based policy, which is more readily deployable in the real-world outside of constrained environments. Can a similar technique be used here for learning visual policies? Moreover, [1] and [2] seem to be highly relevant to this work and must be included in the related works.

2. Motion-planning is classically beneficial for collision-free planning. Have the authors tried any tasks where motion-planning provides any additional benefits such as finding collision-free trajectories in cluttered environments?

3. How can this method be extended to complex manipulation tasks which are non-trivial for motion-planning to solve, such as complex assembly tasks or dynamic object manipulation?

4. While there are several benefits to using classical motion planners, what are the challenges and bottlenecks of using them in terms of both computational complexity as well as real-time efficacy? It would be helpful to know how much time and resources are allocated to collecting expert demonstrations for each of these tasks.

5. It is unclear how the expert trajectories are collected for the collaborative task of stool-lifting with two arms. An elaborate explanation of that would be helpful.

[1] Distilling Motion Planner Augmented Policies into Visual Control Policies for Robot Manipulation.

[2] Motion Planner Augmented Reinforcement Learning for Obstructed Environments.

**Robotics Focus:**

4

**Summary Of Paper:**

This paper proposes a two-stage framework for learning manipulation policies, wherein the first stage, motion-planning is used to generate expert demonstrations removing the need for collecting human-generated expert trajectories. In stage 2, these demonstrations are used to augment the replay buffer of an off-policy deep reinforcement learning for bootstrapping the learning process for various dexterous and whole-body manipulation tasks.

**Summary Of Recommendation:**

The paper is clearly written and easy to follow. While the problem formulation as well as the experiments are convincing, there are several open questions that, if addressed, will help the community to further take this approach to a generalizable and scalable solution.

---

### Official Review · Reviewer_YCCA · 2024-07-21

**Originality:** 2
**Technical Quality:** 3
**Clarity Of Presentation:** 3
**Potential Impact:** 2
**Recommendation:** 3
**Confidence:** 4

**Review:**

**Strengths**
1. The high-level motivation of the paper is well taken. I agree that model-based planners can play a big role in guiding reinforcement learning agents. While we have seen good results from using human-generated demonstrations to guide RL, model-based planners can certainly play a bigger role.
2. The paper is reasonably clearly written.
3. The investigation of different ways to combine planner generated demos with RL generated transitions is informative.
4. Unsurprisingly, the proposed pipeline leads to significantly better learning on a number of interesting manipulation tasks. I particularly like the transfer of policies to real robots.

**Weaknesses/Comments**
1. My biggest concern with this paper is that it does not compare the proposed manipulation planner with any baseline whatsoever. While I understand that many manipulation planners make simplifying assumptions about the system dynamics, it is still worthwhile to compare with them. For example, I believe some of the tasks, such as, box-grasp and stool-lift, could be solved even with a quasi-static assumption. An experimental comparison could better buttress their point about the need to plan with full dynamics.
2. This paper misses highly relevant prior work that do or could potentially generate dynamically feasible trajectories (see [1] and [2]). The proposed planner seems similar to these methods in that it combines tree search with a set of actions generated heuristically or using trajectory optimization. Hence, I am not sure about its novelty and significance.
3. The planner has no theoretical properties of completeness or suboptimality.

[1] Natarajan, Ramkumar, et al. "Torque-limited manipulation planning through contact by interleaving graph search and trajectory optimization." 2023 IEEE International Conference on Robotics and Automation (ICRA). IEEE, 2023.

[2] Cheng, Xianyi, et al. "Enhancing dexterity in robotic manipulation via hierarchical contact exploration." IEEE Robotics and Automation Letters 9.1 (2023): 390-397.

**Quality Of The Limitations Section:**

3

**Questions For Rebuttal:**

1. Please experimentally compare the proposed planner with baselines, even if they make the quasi-static or quasi-dynamic assumption. For example, the authors have cited [14] and [15].
1. Please clarify how the planner compares with and differs from [1] and [2] (see review). Ideally I would like to see an experimental comparison but a discussion may suffice if the authors can clearly articulate how their method is superior.

**Robotics Focus:**

4

**Summary Of Paper:**

This paper proposes a flexible and general manipulation planner that can generate dynamically feasible trajectories for dexterous and whole body manipulation. The planner is used to generate demonstrations for guiding a reinforcement learning agent in simulation. The learned manipulation policy is then deployed zero-shot on a real robot system. The overall pipeline is evaluated on manipulation tasks showing both efficient learning and reasonable sim-to-real transfer.

**Summary Of Recommendation:**

The paper makes two contributions: development of a model-based manipulation planner and using the planner to guide an RL agent. I am unable to recommend acceptance because I am not convinced of the novelty and significance of the planner due to a lack of baselines.

---

### Official Review · Reviewer_feZG · 2024-07-23
**interesting paper**

**Originality:** 2
**Technical Quality:** 3
**Clarity Of Presentation:** 3
**Potential Impact:** 2
**Recommendation:** 3
**Confidence:** 3

**Review:**

Overall, this is an interesting paper with good demonstrations, with interesting sim2real experiments.

 In general, the proposed method relies on search which seems to require quite a bit of front loading effort but it can be applied in a variety of tasks like performing in-hand manipulation and lifting a stool using two spot robots. The front-loaded efforts include quite a bit of hand-crafted engineering for constructing the rewards (distance, proximity, reachability), node selection criteria (i.e. possible actions). Then, further implementation is needed to boostrap these demonstrations for RL. These learned policies can even be transferred from sim2real, which is nice, however, these are mainly for rigid objects for which there might be less sim2real gap. Still, performing tasks like lifting a stool with two spots (which might be quite challenging to teleop for imitation learning) is still great.

The major limitation, as the authors point out, is the need for mocap for observing the full state of the objects, which is not ideal for real-world scenarios. There also seems to be missing comparison to existing state-of-art algorithms.

**Quality Of The Limitations Section:**

3

**Questions For Rebuttal:**

- Can you compare to other previous state of art algorithms? Or some way to show improvement of your method beyond existing state of art algorithms? This seems possible in sim, potentially.

**Robotics Focus:**

4

**Summary Of Paper:**

the paper propose a planner for boostrapping RL for manipulation tasks

**Summary Of Recommendation:**

strong real-world experiments and the technical details seem sound

---

### Official Review · Reviewer_tB8p · 2024-07-29
**Good paper to read, but the novelty of the paper needs to be justified**

**Originality:** 2
**Technical Quality:** 3
**Clarity Of Presentation:** 4
**Potential Impact:** 3
**Recommendation:** 2
**Confidence:** 4

**Review:**

Strength
1. The paper is easy to read, and Figure 1 helps the reader to understand the proposed method.
2. The proposed method makes sense in the sense that one can pre-train policies with a handful of trajectory data from the planner. And the planner here is another form of demonstration.

Weakness
1. One concern is that the RL policies' performance is bounded by the performance of the planner. In the paper, the authors do not compare the quality of the resulting trajectories to other methods that solve the optimization problem of the planner. It would be nice to comment on the quality of the trajectory and its effect on the RL policies.
2.  Another issue is that if we can use a reliable planner and solver to get a good trajectory, why not just deploy such a method in the real world? And the benefit of neural networks is the ability to generalize the unseen setting. I think the experiment section does not have experiments in which you show the planner solver fails to find a solution while the RL policies do not. It would be nice to hear more thoughts on this from the authors.
3. Using tree-search to find solver solutions is not new -- it could be traced back to MCTS or Alpha-Go-like system. Recently, there have been papers that propagate the gradient to the entire solver. As a result, the seem to lack novelty.

**Quality Of The Limitations Section:**

3

**Questions For Rebuttal:**

1. Could authors give an example of how fast the proposed method can run (wall clock time) under what dimension of the states and actions?

**Robotics Focus:**

4

**Summary Of Paper:**

The motivation of this paper is that collecting human demonstration data is costly and not scalable. One would like to use motion planners to generate some demonstration data. However, the motion planner often runs under the assumption of quasi-static dynamics, and the resulting demonstration data might lack action labels. To address this challenge, the paper proposed to use the gradient descent method that computes the gradient through the planner. The resulting generated data is used to train control policies. The goal is not to develop a method that improves the solution-finding capability of the planner but to use that dataset to bootstrap the learning policy.  To find the solution given the dynamics or simulation of the robot, the paper proposes to use tree search methods to find a good planner solution for the whole-body manipulation. The process goes through (1) goal sampling, (2) reward update, (3) node selection, (4) action selection, and (5) transit to the next state given action and the current state.   The second half of the paper talks about how to use such data to train RL policies. The first approach is to add data to the replay buffer, and the second approach is to pre-train the Q function on such an offline dataset.  Finally, the proposed approach is tested in real-world robot tasks.

**Summary Of Recommendation:**

For the above weakness, the paper is on the borderline.

---

### Author Rebuttal · Authors · 2024-08-09

Dear area chair and reviewers

Please find the rebuttal material in the zip folder which contains our update manuscript (changes in red), a document with the newly created plots and tables, and a video of the new results.

---

### Decision · Program_Chairs · 2024-09-05

**Decision:**

Accept

**Comment:**

Strengths: - This work integrates motion planners with RL to bootstrap learning and create an efficient learning approach that can be used in diverse applications like in-hand manipulation and lifting with robots. - Sim2real transfer of learned policies Weaknesses: - Lack of baselines: Authors must provide comparisons against baselines from literature including some mentioned by reviewers. How does the performance differ from a pure motion planning approach? Reviewers also raise concerns about the overall novelty of the proposed approach which could be addressed through writing and baseline comparisons too. - The heuristics used to get good demonstrations from the planner make the generalizability of the approach concerning. Authors should provide more motivation for why this is scalable to more tasks especially tasks that might be challenging for motion planning? - Dependence on state space in the real-world is constraining. How does performance deteriorate when using visual policies?

Post-rebuttal discussion: The key points addressed in the rebuttal are: - Baseline comparisons and novelty: Authors have added comparisons with other approaches from literature and added a new dynamic throwing task in simulation. They have also added discussion about their approach versus others from literature aiming to better highlight their novelty. However the additional simulation comparison experiments are too simplistic and the paper could benefit from stronger baseline comparisons especially in the real-world, which we encourage the authors to include in the camera-ready version.